# Locus coeruleus to basolateral amygdala noradrenergic projections promote anxiety-like behavior

**Jordan G McCall**[1,2,3,4*†‡§¶], **Edward R Siuda**[1,2,3,4†**], **Dionnet L Bhatti**[1,4], **Lamley A Lawson**[1], **Zoe A McElligott**[5,6], **Garret D Stuber**[5,7,8], **Michael R Bruchas**[1,2,3,4,9*]

[1]Department of Anesthesiology, Division of Basic Research, Washington University School of Medicine, St. Louis, United States; [2]Washington University Pain Center, Washington University School of Medicine, St. Louis, United States; [3]Department of Neuroscience, Washington University School of Medicine, St. Louis, United States; [4]Division of Biology and Biomedical Sciences, Washington University School of Medicine, St. Louis, United States; [5]Department of Psychiatry, University of North Carolina, Chapel Hill, United States; [6]Bowles Center for Alcohol Studies, University of North Carolina, Chapel Hill, United States; [7]Department of Cell Biology and Physiology, University of North Carolina, Chapel Hill, United States; [8]Neuroscience Center, University of North Carolina, Chapel Hill, United States; [9]Department of Biomedical Engineering, Washington University, St. Louis, United States

**\*For correspondence:**
jordangmccall@wustl.edu (JGM);
bruchasm@wustl.edu (MRB)

[†]These authors contributed equally to this work

**Present address:** [‡]Department of Anesthesiology, Washington University, St. Louis, United States; [§]Department of Pharmaceutical and Administrative Sciences, St. Louis College of Pharmacy, St. Louis, United States; [¶]Center for Clinical Pharmacology, Washington University School of Medicine and St. Louis College of Pharmacy, St. Louis, United States; [**]Trevena Inc., King of Prussia, United States

**Abstract** Increased tonic activity of locus coeruleus noradrenergic (LC-NE) neurons induces anxiety-like and aversive behavior. While some information is known about the afferent circuitry that endogenously drives this neural activity and behavior, the downstream receptors and anatomical projections that mediate these acute risk aversive behavioral states via the LC-NE system remain unresolved. Here we use a combination of retrograde tracing, fast-scan cyclic voltammetry, electrophysiology, and in vivo optogenetics with localized pharmacology to identify neural substrates downstream of increased tonic LC-NE activity in mice. We demonstrate that photostimulation of LC-NE fibers in the BLA evokes norepinephrine release in the basolateral amygdala (BLA), alters BLA neuronal activity, conditions aversion, and increases anxiety-like behavior. Additionally, we report that $\beta$-adrenergic receptors mediate the anxiety-like phenotype of increased NE release in the BLA. These studies begin to illustrate how the complex efferent system of the LC-NE system selectively mediates behavior through distinct receptor and projection-selective mechanisms.

## Introduction

The locus coeruleus noradrenergic system (LC-NE) comprises a widespread projection network throughout the central nervous system capable of modulating a diverse range of behaviors including arousal, learning, pain modulation, and stress-induced negative affective states (**Berridge and Waterhouse, 2003**; **Sara, 2009**). Understanding the neural circuit basis for how this nearly ubiquitous neuromodulatory network exerts influence on negative affect is a critical step towards therapeutically targeting stress-induced neuropsychiatric disorders (**Schwarz and Luo, 2015**; **Schwarz et al., 2015**; **Reyes et al., 2015**; **McCall et al., 2015**; **Arnsten et al., 2015**; **Kebschull et al., 2016**). One particular efferent projection from the LC is to the basolateral amygdala (BLA). The BLA is an

important candidate anatomical substrate for the widely known role of norepinephrine (NE) in affective behaviors (*Berridge and Waterhouse, 2003*; *Schwarz et al., 2015*; *Davis, 1992*; *Valentino and Aston-Jones, 2010*; *Robertson et al., 2016*, *2013*; *Grissom and Bhatnagar, 2011*; *Siuda et al., 2015a*, *2016*; *Plummer et al., 2015*). The BLA is notable for integrating sensory information to encode and drive diverse and opposing affective behaviors including anxiety, fear, aversive, and reward behaviors (*Kim et al., 2013*; *Stuber et al., 2011*; *Tye et al., 2011*; *Namburi et al., 2015*; *Gore et al., 2015*; *Belova et al., 2007*; *Beyeler et al., 2016*; *Wolff et al., 2014*; *Bermudez and Schultz, 2010*; *Bruchas et al., 2009*; *Knoll et al., 2011*; *Sugase-Miyamoto and Richmond, 2005*; *Crowley et al., 2016*; *Sears et al., 2013*; *Roozendaal et al., 2008*, *2006*; *Miranda et al., 2007*). Notable efforts to uncover the role of the BLA and adrenergic signaling in consolidation of fear memories have been reported (*Sears et al., 2013*; *Roozendaal et al., 2008*, *2006*), as well as recent studies showing that acute stress activates BLA adrenergic receptors (agnostic to the source of NE) to promote anxiety and other stress-related behaviors (*Miranda et al., 2007*; *Buffalari and Grace, 2009a*, *2009b*; *Chang and Grace, 2013*). Similarly acute stress paradigms cause selective activation of LC-NE neurons (*McCall et al., 2015*). Together, there have been significant efforts to examine how source-independent noradrenergic (importantly, there are multiple sources of NE innervating the BLA (*Robertson et al., 2016*; *Plummer et al., 2015*) signaling in the BLA can alter synaptic plasticity, fear encoding, and memory consolidation, yet few studies have directly examined how the neuromodulatory LC-NE system utilizes BLA output to alter acute risk averse behaviors, such as anxiety (*Grissom and Bhatnagar, 2011*; *Buffalari and Grace, 2009a*, *2009b*, *2007*).

Downstream and independent of this projection, recent studies have demonstrated that direct activation of both basolateral amygdala (BLA) cell bodies or their projections is both anxiogenic and socially aversive (*Siuda et al., 2015a*, *2016*; *Tye et al., 2011*; *Felix-Ortiz et al., 2013*, *2016*; *Felix-Ortiz and Tye, 2014*). Furthermore, it has also been demonstrated that increasing BLA excitatory output through Gαs G-protein activation, and more specifically, β-adrenergic receptor signaling causes acute social anxiety (*Siuda et al., 2016*). Separately, noradrenergic cell firing in the LC has been shown to increase in the context of stressful stimuli (*McCall et al., 2015*; *Abercrombie and Jacobs, 1987a*, *1987b*; *Aston-Jones et al., 1999*; *Mana and Grace, 1997*). While the anatomical projections from the LC and their cell types have been studied for several years, the precise mechanisms by which fibers from the LC can directly influence BLA function to promote negative affective behaviors are not understood. Specifically, how LC-BLA projections generate affective behavioral responses through specific receptor systems and modulation of cell activity is unknown.

To determine the role of locus coeruleus noradrenergic influence on BLA function and negative affective behavior we optogenetically manipulated LC-NE inputs into the BLA, directly testing whether NE is released from LC terminals into the BLA and whether this terminal stimulation can drive anxiety-like and aversive behavioral responses. We demonstrate that photostimulation of LC projections to the basolateral amygdala releases NE and that this photostimulation evokes downstream modulation of neuronal activity in BLA neurons that project to anxiogenic brain regions. Stimulation of these fibers is sufficient to produce conditioned aversion and the stimulation-induced increased noradrenergic tone is sufficient to produce anxiety-like behavior mediated by local β-adrenergic receptor activity (β-ARs) in the BLA. Taken together, we report a previously undefined role for LC-BLA projections in mediating negative affective behavior through activation of β-ARs.

## Results

### Genetic and anatomical isolation of BLA-projection LC-NE neurons

Our previous work demonstrated that increased tonic LC-NE activity induces anxiety-like and aversive behavioral responses (*McCall et al., 2015*; *Siuda et al., 2015a*), we next sought to test whether these same behaviors can be generated by stimulating LC-NE fibers at localized projections in the BLA. To examine the potential sources of NE from the LC to the BLA we identified and isolated the projection using two distinct retrograde tracing approaches. First, we injected the tracer Fluorogold into the BLA of wild-type mice (*Figure 1A*). Consistent with previous studies, this non-selective retrograde tracing approach revealed known inputs into the BLA from the LC (*Robertson et al., 2016*; *Asan, 1998*; *Fallon et al., 1978*) (*Figure 1B*). We next used a dual injection strategy to anatomically isolate BLA-projecting LC-NE$^+$ neurons. To do so, we used mice expressing Cre under the promoter

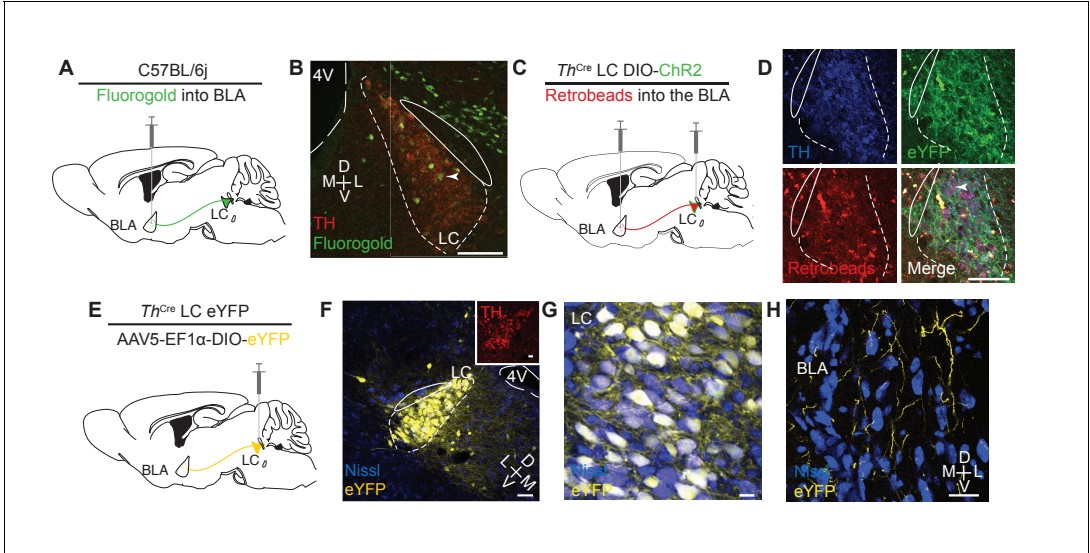

**Figure 1.** Identifying a LC input to the BLA. (**A**) Cartoon depicting fluorgold tracing strategy. (**B**) Representative image (selected from three injected mice) shows robust retrograde labeling of the LC from injection in the BLA (green = pseudocolored Fluorgold, tyrosine hydroxylase = red). Arrowhead indicates example co-localization. Scale bar = 100 μm. 4V = 4th ventricle. The TH⁻ cells dorsal and ventral to the LC are likely part of the medial parabrachial nucleus which has previously identified projections to the BLA (*Saper and Loewy, 1980*). (**C**) Cartoon depicting dual injection tracing strategy for CTB-594 and DIO-ChR2-eYFP. (**D**) Representative images (selected from three injected mice) shows retrograde labeling in LC of red retrobeads and anterograde labeling of TH+ cells (green) (Nissl=blue). Arrowhead indicates example co-localization. Scale bar = 100 μm (**E**) Cartoon depicting anterograde tracing strategy. (**F–H**) Coronal images depict robust eYFP (yellow) labeling in the LC (F and G) and BLA (H) of the same mouse (scale bars = (**F**) 50 μm, (**G**) 10 μm, (**H**) 20 μm. Inset (**F**), tyrosine hydroxylase = red, scale bar = 25 μm.

The following figure supplement is available for figure 1:

**Figure supplement 1.** Further identification of LC input to the BLA.

for tyrosine hydroxylase, the rate-limiting enzyme for catecholamine synthesis (*Th*^IRES-Cre mice) (*Savitt et al., 2005*). Here, we injected a red retrobead tracer into the BLA and the green-labeled adeno-associated virus, AAV5-DIO-Ef1α-ChR2(H134)-eYFP, into the LC (*Th*^IRES-Cre::LC-BLA:ChR2; *Figure 1C and D*). In these animals, we identified BLA-projecting *Th*⁺ LC neurons with the presence of both fluorophores in the same cells (*Figure 1D*). Next, to examine *Th*⁺ terminal innervation of the BLA, we injected the cell-filling, Cre-dependent reporter AAV5-DIO-Ef1α-eYFP in the LC of *Th*^IRES-Cre mice (*Th*^IRES-Cre::LC-BLA:eYFP) (*Figure 1E*). Here we clearly observed labeled LC neurons (*Figure 1F and G*) and their projection fibers terminating in the BLA (*Figure 1H*). To further corroborate these findings, we examined recent projection experiments performed by the Allen Brain Institute Mouse Connectivity (ABIMC) project that also genetically and anatomically isolate this projection (*Oh et al., 2014*). In three different experiments from three different genetic models, we observed LC-BLA projections that are qualitatively similar to our own observations (*Figure 1—figure supplement 1A–L*). We present these findings here for clarity and ease of independent comparison, but this work was performed by ABIMC. Together, these anatomical studies identify a discrete projection of *Th*⁺ neurons from the LC that potentially release endogenous NE into the BLA.

## Optogenetic activation of LC-BLA terminals releases norepinephrine into the BLA

Using the same viral optogenetic strategy as above, we examined whether photostimulation of *Th*^IRES-Cre::LC-BLA:ChR2 projections resulted in NE release at terminals. We validated functional ChR2 expression using whole-cell current clamp recordings of *Th*⁺ LC neurons. As, we previously demonstrated (*McCall et al., 2015*), this targeting method and photostimulation protocol was sufficient to generate action potentials at the LC cell bodies (*Figure 2A*). Next, we used a carbon fiber microelectrode (CFME) to perform fast-scan cyclic voltammetry (FSCV) in the BLA during LC-NE

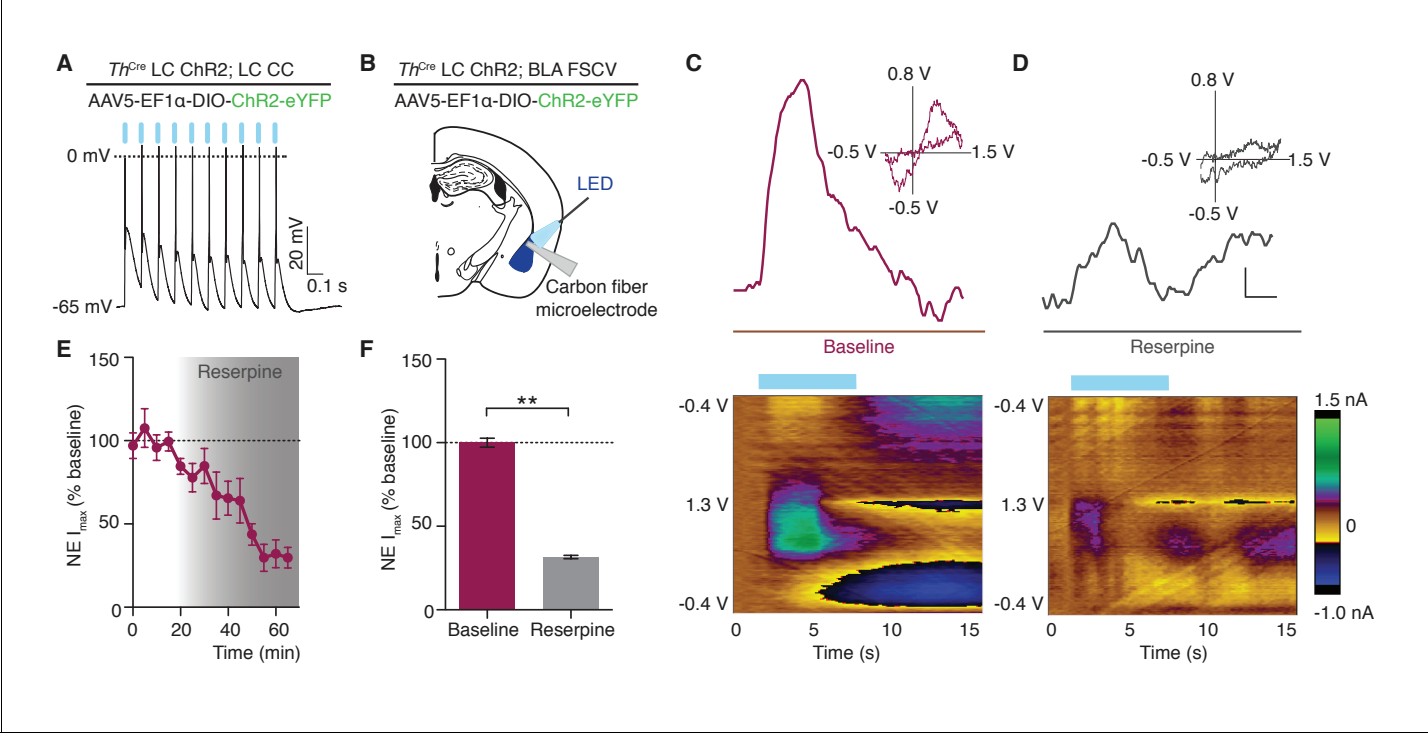

**Figure 2.** Photostimulation of LC terminals in the BLA releases norepinephrine. (**A**) LC neuron firing reliably to 10 Hz optical stimulation (CC=whole cell current clamp). (**B**) Fast scan cyclic voltammetry (FSCV) schematic. (**C–D**) Oxidative and reductive currents (scale bar 2 s by 0.4 nA), with representative cyclic voltammograms (inset) and representative color plots (below) in response to photostimulation are attenuated by reserpine (1 μM). Color plots for baseline and after reserpine (1 μM): Files were collected over 15 s (X-axis) where the carbon fiber microelectrode was ramped with a triangular waveform from −0.4V to 1.3V and back to −0.4V at 400 V/S (Y-axis) and sampled at 10 Hz. 10 Hz, 473 nm blue LED stimulation onset at 2 s. Oxidative currents (nA) are positive in direction and reductive currents are negative (see color coded scale bar on right). (**E**) Attenuation in NE oxidative current in response to reserpine (1 μM) n = 3 pairs; mean ± S.E.M. (**F**) Average of first 20 min and last 15 min in (**E**) (Data represented as mean ± SEM, Paired Student's t-tests to baseline, Mean difference = 68.56, $t_{(2)}$ = 18.75, **p=0.0028, 95% CI [52.82 to 84.29]).

terminal stimulation. Using the extended waveform method (see methods) in acute brain slices of $Th^{IRES-Cre}$::LC-BLA:ChR2 animals, we stimulated slices with 30 5 ms light pulses from a 473 nm LED, at 10 Hz (**Figure 2B**). Photostimulation of BLA slices, produced characteristic cyclic voltammograms and uptake consistent with NE (**McElligott et al., 2013**; **Herr et al., 2012**) ($t_{1/2}$ = 2.0 ± 0.2 s, **Figure 2C and D**). Following a 20 min baseline, 1 μM reserpine (an inhibitor of vesicular monoamine transporters) was perfused on the slices to deplete catecholamines from the axon terminals (**Dahlstroem et al., 1965**). Reserpine treatment significantly attenuated the measured oxidative currents (31.6 ± 1.0% of baseline, **Figure 2E and F**) further confirming optically-evoked catecholamine release in this isolated $Th^{IRES-Cre}$::LC-BLA:ChR2 projection. These findings suggest that optogenetic manipulation of LC-BLA terminals causes release of endogenous NE from the LC into the BLA.

## In vivo photostimulation of LC-BLA terminals modulates BLA activity

BLA neurons have well reported responses to exogenous application of NE (**Buffalari and Grace, 2007**), but their response to endogenous NE release, explicitly from the LC, has not been previously defined. We next determined whether the optically-evoked endogenous catecholamine release would mimic the responses of BLA neurons to exogenous NE. To do so, we examined BLA single-unit activity in $Th^{IRES-Cre}$::LC-BLA:ChR2 mice using 16-channel microelectrode arrays coupled to a fiber optic implant (optrode arrays) (**McCall et al., 2015**; **Sparta et al., 2011**). These optrode arrays were used to isolate and record BLA single-unit activity before, during, and after photostimulation of $Th^{IRES-Cre}$::LC-BLA:ChR2 projections (473 nm, 5 Hz, 10 ms pulse width) (**Figure 3A and B**). In these experiments photostimulation of $Th^{IRES-Cre}$::LC-BLA:ChR2 terminals caused an increase in firing

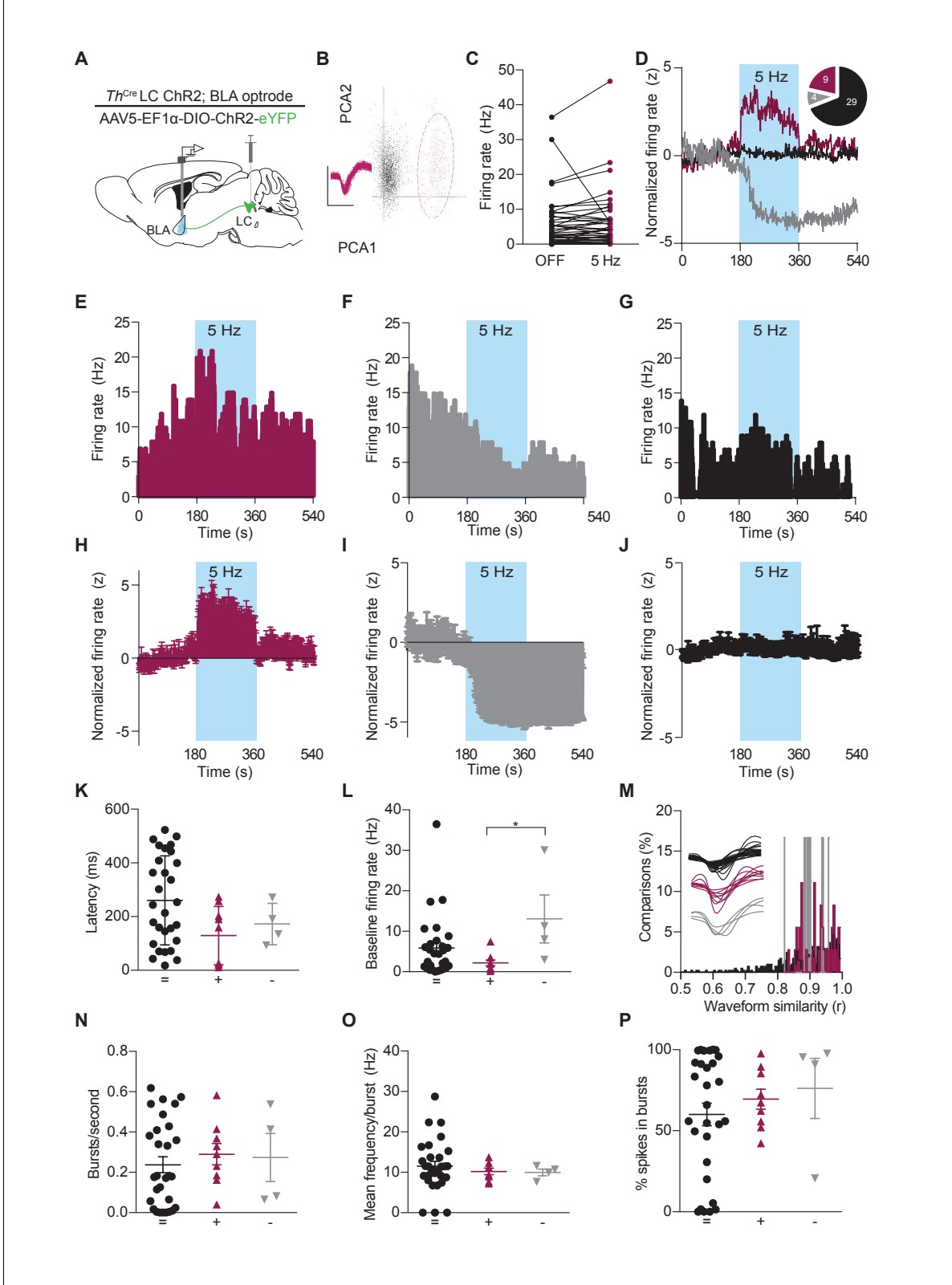

**Figure 3.** Photostimulation of LC terminals in the BLA alters neuronal activity. (**A**) Schematic illustrating single-unit extracellular recording paradigm of BLA neurons modulated by ChR2-expressing LC-BLA terminals. (**B**) Representative principal component analysis plot showing the first two principal components with clear clustering of a single unit (maroon) from the noise (grey). Inset shows the waveform and spikes making up the isolated unit. Y-scale is 150 microvolts and x-scale is 500 ms. (**C**) Recordings from eight hemispheres of six *Th*-Cre[LC-BLA:ChR2] mice show the distribution of firing rates

*Figure 3 continued on next page*

*Figure 3 continued*

present in BLA neurons prior to and following LC-BLA terminal photostimulation (473 nm, 5 Hz, 3 min). (**D**) Average normalized firing rate of neurons that increase (maroon), decrease (grey), or do not change (black) firing rate in response to photostimulation. Inset, shows number of neurons in each group. Representative histograms (1 s bins) of isolated single-units showing increase (**E**) or decrease (**F**), or no change (**G**) in neuronal firing in response to photostimulation (473 nm, 5 Hz, 3 min). Z-scored population responses of neurons showing increase (**H**) or decrease (**I**), or no change (**J**) in neuronal firing in response to photostimulation. (**K**) Response latency following onset of photostimulation for cells that did not alter firing (=) (n = 29), increased firing (+) (n = 9), or decreased firing (-) (n = 4). (Data represented as mean ± SD). (**L**) The same cells sorted by baseline firing rate. (Data represented as mean ± SEM. Kruskal-Wallis test one-way ANOVA for non-parametric data with Dunn's multiple comparisons test, Kruskal-Wallis statistic = 6.536, p=0.0381; + vs. – Mean rank difference = −18.75, adjusted *p=0.0329; + vs. + Mean rank difference = −6.828, adjusted p=0.4341; - vs. = Mean rank difference = 11.92, adjusted p=0.2053.) (**M**) Waveform similarity, within group distribution of linear correlations. Inset, every average waveform for each recorded unit separated by response profile (= black, + maroon, - grey). (**N–P**) Bursting profiles for each recorded neuron. (**N**) Number of bursts per second. (Data represented as mean ± SEM). (**O**) Mean firing rate within bursts for each neuron. (Data represented as mean ± SEM). (**P**) Proportion of recorded spikes that occurred during bursts. (Data represented as mean ± SEM).

The following figure supplement is available for figure 3:

**Figure supplement 1.** Photostimulation of LC terminals in the BLA alters neuronal activity.

frequency in 21.4% of units recorded in the BLA (*Figure 3C–E,H*; *Figure 3—figure supplement 1A*), while some cells (9.5%) displayed inhibitory responses (*Figure 3D,F,I*; *Figure 3—figure supplement 1B*). The remaining neurons (69.0%) appeared to not change in response to photostimulation (*Figure 3D,G,J*; *Figure 3—figure supplement 1C*). Furthermore, in a subset of cells blindly selected (without knowledge of the increase/decrease/static response to photostimulation) following the 5 Hz recordings, we also observed similar increases in firing rates to constant photostimulation, where the overall population of neurons increased firing during stimulation (*Figure 3—figure supplement 1D and E*). In cases where the firing rate increased, the mean ± SD latency to fire following each light pulse was 129.3 ± 108.9 ms, suggesting that this change is not due to fast, monosynaptic neurotransmission, but likely a polysynaptic and/or neuromodulatory effect (*Figure 3K*). Likewise, in cases where the firing rate decreased the mean latency was slower at 172.6 ± 77.01 ms, and neither case was significantly different to neurons whose firing rate did not change following photostimulation (260.4 ± 165.7) (*Figure 3K*).

We next sought to determine whether these differences in neuronal response arise from distinct subsets of BLA neurons. To do so, we examined the baseline firing rate, waveforms, and bursting properties of recorded units. Neurons that increase firing in response to LC-BA terminal stimulation have a significantly lower mean basal firing rate (2.194 ± 2.22 SD) than those that decrease firing (13.06 ± 11.82 SD), however neither group is distinguishable from non-responsive neurons (*Figure 3L*). To quantifiably assess the waveform shapes of the recorded neurons, we calculated the average waveform for each neuron and performed a linear correlation on these values within each group of neurons. These analyses demonstrate that all neurons that increase firing to LC-BLA photostimulation have an r value greater than 0.835 and all neurons that decrease have an r value above 0.820, while the population of neurons that did not significantly respond to photostimulation have an r above 0.515. These results indicate that neurons that do not respond to photostimulation are less internally similar, while the excited and inhibited cells are more similar within groups, suggesting that these modulated neurons are more likely to each be part of single class of neurons (*Figure 3M*). Finally, we examined the bursting properties of the recorded neurons. While most neurons exhibited some bursting properties, no differences were found between groups in terms of frequency of bursting (*Figure 3N*), mean firing rate during bursts (*Figure 3O*), or the percentage of spikes from the recording sessions that occurred within a burst (*Figure 3P*). Together, these heterogeneous firing properties are consistent with previous studies using iontophoresis of NE into the BLA, and further highlight the complex pharmacological and circuit activity that NE modulates within the BLA (*Buffalari and Grace, 2007*; *Ferry et al., 1997*; *Huang et al., 1996*). These results suggest that photostimulation of LC terminals in the BLA causes varied neuronal firing rate responses, with the majority of photostimulation-responsive neurons increasing firing in response to the LC-NE terminal photostimulation.

## LC-BLA terminal stimulation biases activation towards anxiety-promoting BLA neurons

The BLA is thought to be a heterogeneous hub for emotional processing containing separable populations for regulating either positive or negative affect. Recent studies have suggested that these opposing populations are distinct in either their projection target or their cell-type (*Stuber et al., 2011*; *Namburi et al., 2015*; *Beyeler et al., 2016*; *Felix-Ortiz et al., 2016*; *Burgos-Robles et al., 2017*; *Kim et al., 2016*, *2017*; *Correia et al., 2016*). Given the role of BLA adrenergic receptors in modulating anxiety-like and aversion behaviors (*Siuda et al., 2015a*, *2016*), we hypothesized that LC-NE innervation of BLA neurons may preferentially bias activation of neurons that promote anxiety-like behavior such as the ventral hippocampus (vHPC)- and central amygdala (CeA)- projecting BLA neurons as opposed to projections that promote positive affect and anxiolysis such as nucleus accumbens (NAc)-projecting BLA neurons. Using a combination of retrograde viral tracing, immuno-histochemistry, and optogenetic stimulation in *Th*-Cre$^{\text{LC-BLA:ChR2}}$ mice, we assessed these potential tri-synaptic circuits that may underlie anxiety-like behaviors (*Figure 4A*). Photostimulation of LC-NE terminals in the BLA (5 Hz, 10 ms) significantly increases the number of cFos-expressing BLA neurons in *Th*$^{\text{IRES-Cre}}$::LC-BLA:ChR2 animals compared to ChR2-lacking (*Figure 4B*; *Figure 4—figure supplement 1A–D*) and contralateral BLA controls following photostimulation (*Figure 4—figure supplement 1E*). To assess whether the cFos+ BLA neurons resulting from LC-NE terminal activation were biased toward a particular class of BLA projection neurons we next repeated the experiment with the retrograde tracer Cholera toxin subunit B (CTB) injected into BLA projection regions (vHPC, CeA, and NAc). These immunohistochemistry studies reveal that the cFos present following LC-NE terminal activation in the BLA, overlaps significantly more with vHPC- and CeA-targeted compared to NAc-targeted CTB in the BLA (*Figure 4C–G*; *Figure 4—figure supplement 1F–H*). These results suggest that LC-NE terminals in the BLA preferentially activate vHPC- and CeA- projecting BLA neurons thought to be involved in modulating negative valence and affective behaviors.

## LC-BLA terminal stimulation is sufficient to elicit a conditioned place aversion

After identifying the LC terminals to the BLA as a projecting source of NE, that photostimulation of these terminals are capable of altering BLA activity, and that this photostimulation appears to bias BLA activation towards circuits involved in negative affect, we next tested whether this same pattern of terminal stimulation can drive an immediate negative valence. Fiber optics were implanted at LC-NE terminal sites in the BLA of *Th*$^{\text{IRES-Cre}}$ mice with AAV5-DIO-Ef1α-ChR2(H134)-eYFP previously injected into the LC (*Th*$^{\text{IRES-Cre}}$::LC-BLA:ChR2) (*Figure 5A*, *Figure 5—figure supplement 1A*). Surprisingly, photostimulation of *Th*$^{\text{IRES-Cre}}$::LC-BLA:ChR2 terminals using the parameters that induce a real-time place aversion at LC-NE cell bodies (*McCall et al., 2015*) failed to elicit a real-time aversion in *Th*$^{\text{IRES-Cre}}$::LC-BLA:ChR2 animals (*Figure 5B and C*). There was no significant change in place testing behavior (*Figure 5C*) or locomotor behavior (*Figure 5D*) within animals or compared to *Th*$^{\text{IRES-Cre}}$::LC-BLA:eYFP controls at either a physiologically-relevant 5 Hz or an over-driven 60 Hz photostimulation. This finding is particularly interesting in light of previous systemic pharmacology experiments (*McCall et al., 2015*) that suggested the immediate negative valence induced by LC-NE stimulation is mediated by α$_1$ adrenergic receptor activation. These results indicate that the real-time negative valence of LC-NE stimulation is likely produced via another LC-NE projection and one that is not recruited through antidromic stimulation of BLA-projecting LC neurons.

It is also possible that the neuromodulatory effects on this circuit require a long-term associative memory component compared to fast-acting, acute activation of similar pathways (*Kim et al., 2013*; *Namburi et al., 2015*; *Haubensak et al., 2010*). To directly test whether long-term learned associations are produced by *Th*$^{\text{IRES-Cre}}$::LC-BLA:ChR2 stimulation, we employed a Pavlovian conditioned place aversion (CPA) assay similar to our previous work (*McCall et al., 2015*; *Siuda et al., 2015a*; *Al-Hasani et al., 2013*; *Siuda et al., 2015b*) (*Figure 5E*, *Figure 5—figure supplement 1B*). When allowed to freely explore two contextually-differentiated chambers, following two conditioning days of photostimulation of *Th*$^{\text{IRES-Cre}}$::LC-BLA:ChR2 terminals (5 Hz, 10 ms), animals expressing ChR2-eYFP spent significantly less time in the context that was previously paired with photostimulation compared to the *Th*$^{\text{IRES-Cre}}$::LC-BLA:eYFP controls (*Figure 5F and G*). Neither group of mice showed significant changes in locomotor activity before, during, or after conditioning (*Figure 5H*). Together,

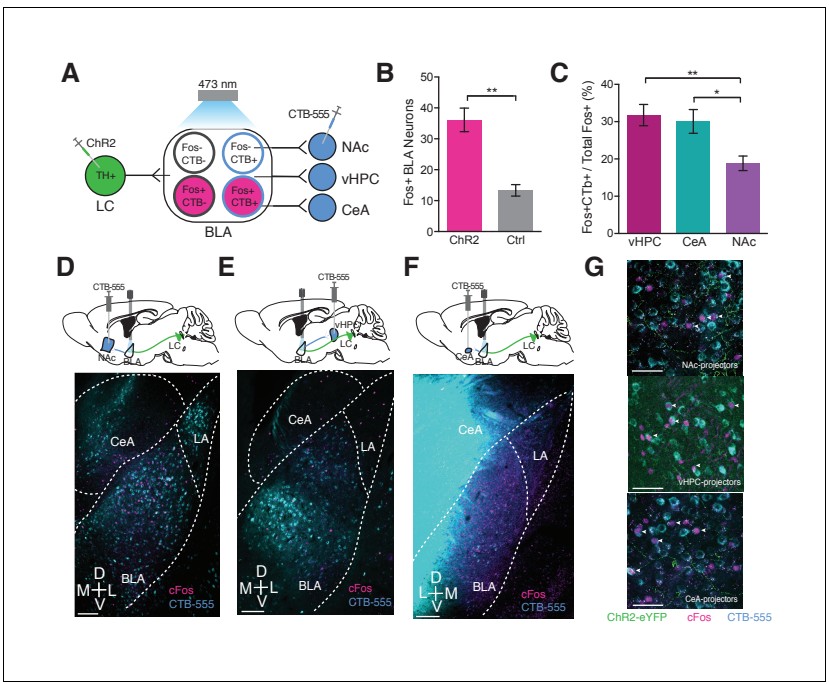

**Figure 4.** Photostimulation of LC terminals in the BLA preferentially activates BLA circuitry associated with anxiety-like behavior. (A) Diagram of viral and optogenetic strategy. (B) 5 Hz photostimulation increases cFos expression within the BLA in $Th^{IRES-Cre}$::LC-BLA:ChR2$^+$ animals compared to $Th^{IRES-Cre}$::LC-BLA:ChR2$^-$ controls (Data represented as mean ± SEM, n = 9 ChR2, n = 4 Ctrl; average of 3 sections/mouse; Student's t-test, Mean difference = 19.17, $t_{(10)}$ = 4.005, \*\*p=0.0040, 95% CI [−35.47 to −10.11]. (C) 5 Hz photostimulation increases cFos expression significantly more in BLA neurons projecting to the vHPC and CeA compared to NAc in $Th^{IRES-Cre}$::LC-BLA:ChR2 animals (Data represented as mean ± SEM, n = 9 vHPC$^{CTB}$, n = 6 CeA$^{CTB}$, n = 9 CeA$^{CTB}$, 3 sections per mouse; One-Way ANOVA, Bonferroni's Multiple Comparison Test, $F_{2,20}$ = 7.199, \*\*p=0.0044; $Th^{IRES-Cre}$::LC-BLA: ChR2:CTB-vHPC vs. $Th^{IRES-Cre}$::LC-BLA:ChR2:CTB-NAc Mean difference = 12.95, $t_{(20)}$ = 3.585, \*\*p<0.01 95% CI [3.511 to 22.39]; $Th^{IRES-Cre}$::LC-BLA:ChR2:CTB-CeA vs. $Th^{IRES-Cre}$::LC-BLA:ChR2:CTB-NAc. Mean difference = 11.25, $t_{(20)}$ = 2.802, \*p<0.05 95% CI [0.7605 to 21.74]. Representative images of the BLA expressing cFos after 5 Hz photostimulation in (D) $Th^{IRES-Cre}$::LC-BLA:ChR2 injected with CTB in NAc, (E) vHPC, or (F) CeA. Scale bar, 100 μm. (G) Confocal images showing colocalization of CTB and cFos after 5 Hz photostimulation in the NAc, vHPC, and CeA. Scale bar, 50 μm.

The following figure supplement is available for figure 4:

**Figure supplement 1.** Photostimulation of LC terminals in the BLA preferentially activates BLA circuitry associated with negative affect.

these results suggest that there is a learned negative association produced by photostimulation of $Th^+$ fibers in the BLA. This association is capable of producing avoidance from contexts in which the photostimulation previously occurred.

## Optogenetic targeting of LC-NE projections to the BLA promotes anxiety-like behavior through beta-adrenergic receptor activity

Following the observation that stimulation of LC-BLA fibers alone is sufficient to cause a conditioned aversion, we next tested whether the same stimulation of these terminals would drive acute anxiety-like behaviors similar to those we recently reported in response to direct LC-NE cell body activation. Photostimulation of $Th^{IRES-Cre}$::LC-BLA:ChR2 terminals using parameters that induce anxiety-like behavior at LC-NE cell bodies (*McCall et al., 2015*) resulted in a significant decrease in time spent in the center of the open field (OFT) with no significant change in general locomotor activity (*Figure 6A–D*, *Figure 6—figure supplement 1A*). These experiments suggest that stimulation of only the subset of LC-NE fibers that project to the BLA is sufficient to induce anxiety-like behaviors.

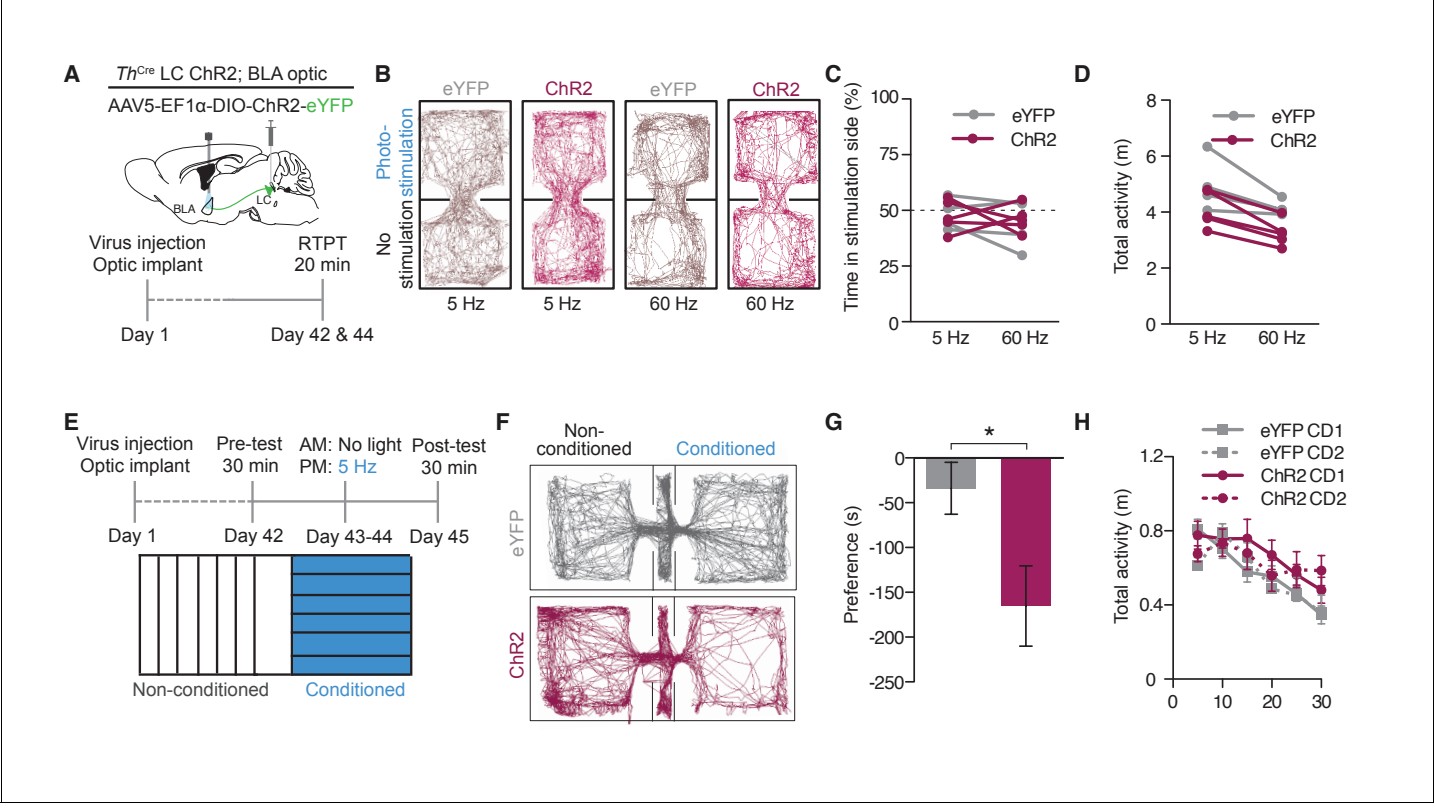

**Figure 5.** Photostimulation of LC terminals in the BLA causes conditioned aversion. (**A**) Cartoon of viral and fiber optic delivery strategy and calendar of real-time place testing studies. (**B**) Representative traces of behavior at different frequencies. (**C**) Frequency response of RTPT and (**D**) locomotor activity at 5 and 60 Hz. Data represented as mean ± SEM, n = 4 eYFP, 5 ChR2. (**E**) Conditioned place aversion (CPA) behavioral calendar. (**F**) Representative CPA traces. (**G**) $Th^{IRES-Cre}$::LC-BLA:ChR2 mice (maroon; n = 9) show a conditioned aversive response to chamber paired with photostimulation compared to $Th^{IRES-Cre}$::LC-BLA:eYFP controls (grey; n = 8) (Data represented as mean ± SEM, Student's t-test, Mean difference = 131.3, $t_{(15)}$ = 2.39, *p=0.0303, 95% CI [14.32 to 248.4) following two days of conditioning with (**H**) no significant differences in locomotor behavior during conditioning day 1 (CD1) and or conditioning day 2 (CD2) in CPA.

The following figure supplement is available for figure 5:

**Figure supplement 1.** Photostimulation of LC terminals in the BLA drive aversive behavior.

Our prior work revealed that systemic antagonism of β-ARs blocks the anxiogenic, but not the real-time aversive, components of tonic LC-NE stimulation (*McCall et al., 2015*). We next tested whether β-AR activity in the BLA is necessary for LC-NE mediated anxiogenesis. To do so we implanted a combined fiberoptic-fluid cannula into the BLA and delivered either an artificial cerebrospinal fluid vehicle or the non-selective β-AR antagonist, Propranolol (1 μg)(*Roozendaal et al., 2008*, *2006*; *Chang and Grace, 2013*; *Shanks et al., 1966*), prior to photostimulation (*Figure 6E–H*, *Figure 6—figure supplement 1B*). The same photostimulation paradigm that was anxiogenic in the OFT, caused a decrease in time spent in the open area of the elevated zero maze (EZM) of ChR2⁺ animals that received vehicle, compared to YFP-expressing, vehicle controls (*Figure 6G and H*). Importantly, when we locally antagonized β-ARs directly in the BLA prior to photostimulation, anxiogenesis was completely blocked in ChR2⁺ animals with no effect on eYFP-expressing, propranolol controls (*Figure 6G and H*). These findings suggest that local β-AR activation by NE release from LC-BLA terminals is the substrate responsible for the photostimulation-induced anxiety-like behavior.

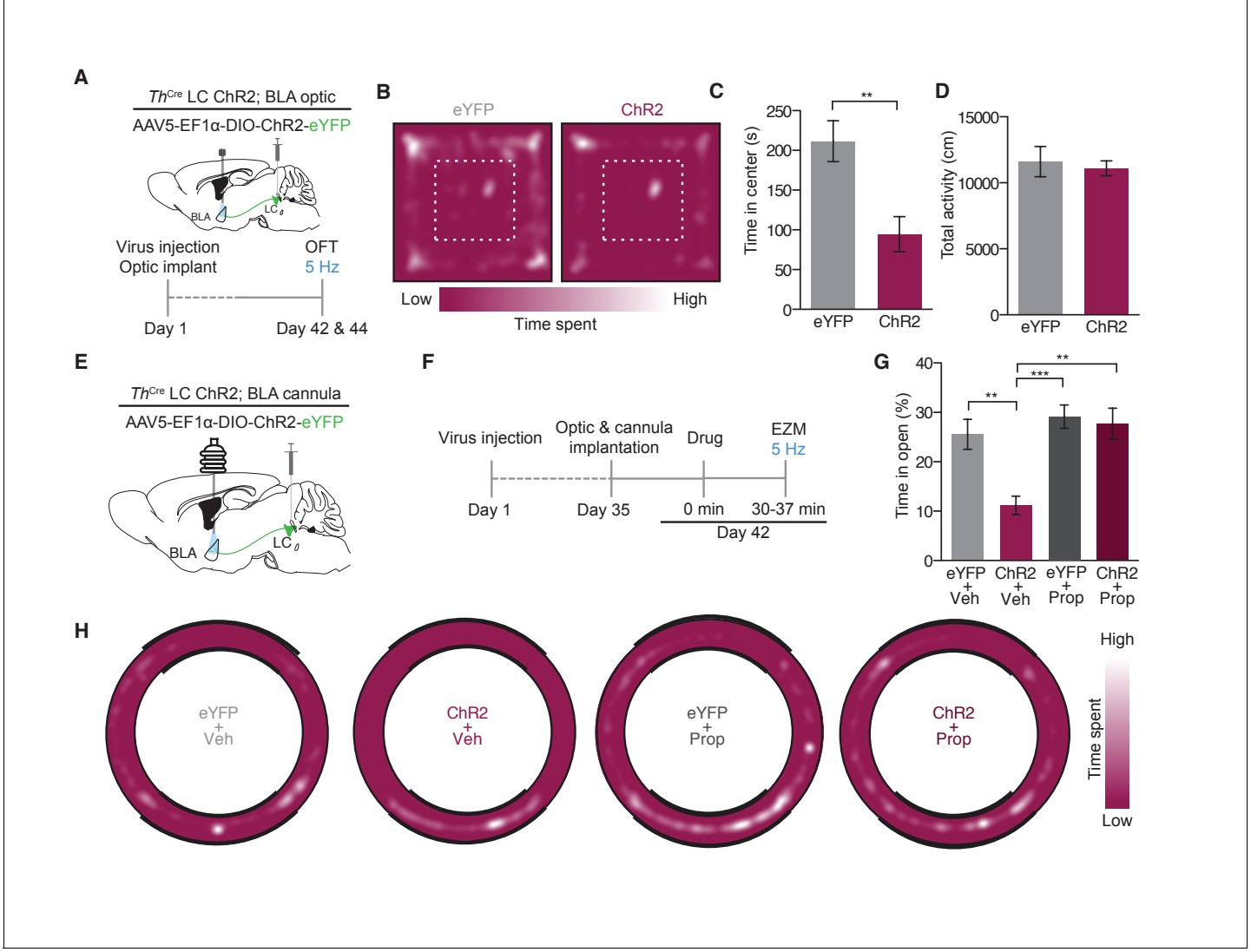

**Figure 6.** Photostimulation of LC terminals in the BLA promotes anxiety-like behavior through beta-adrenergic receptors. (**A**) Calendar of OFT studies. (**B**) Representative heat maps of activity during OFT. (**C**) 5 Hz photostimulation causes an anxiety-like phenotype in OFT of $Th^{IRES-Cre}$::LC-BLA:ChR2 animals compared to $Th^{IRES-Cre}$::LC-BLA:eYFP controls (Data represented as mean ± SEM, n = 10 eYFP, 11 ChR2; Student's t-test, Mean difference = 116.9, $t_{(19)}$ = 3.46, **p=0.0026, 95% CI [46.20 to 187.5] with (**D**) no change in locomotor activity (Data represented as mean ± SEM). (**E**) Cartoon of viral, cannula, and fiber optic delivery strategy and (**F**) calendar of EZM behavior. (**G**) 5 Hz photostimulation causes an anxiety-like phenotype in EZM of $Th^{IRES-Cre}$::LC-BLA:ChR2 animals compared to $Th^{IRES-Cre}$::LC-BLA:eYFP controls, which is reversed by intra-BLA propranolol pretreatment (Data represented as mean ± SEM, n = 11 eYFP + Vehicle, n = 9 ChR2 + Vehicle, n = 7 eYFP + Propranolol, n = 8 ChR2 + Propranolol; One-Way ANOVA, Bonferroni's Multiple Comparison Test, $F_{3,31}$ = 8.95, p=0.0002; $Th^{IRES-Cre}$::LC-BLA:eYFP+Vehicle vs. $Th^{IRES-Cre}$::LC-BLA:ChR2+Vehicle. Mean difference = 14.74, $t_{(2)}$ = 3.94, **p<0.01 95% CI [4.21 to 25.27]; $Th^{IRES-Cre}$::LC-BLA:ChR2+Vehicle vs. $Th^{IRES-Cre}$::LC-BLA:ChR2+Propranolol. Mean difference = 16.93, $t_{(2)}$ = 4.19, **p<0.01 95% CI [5.46 to 28.31]; $Th^{IRES-Cre}$::LC-BLA:eYFP+Propranolol vs. $Th^{IRES-Cre}$::LC-BLA:ChR2+Vehicle. Mean difference = 18.37, $t_{(2)}$ = 4.39, ***p<0.001 95% CI [6.57 to 30.18]. (**H**) Representative heat maps of activity during EZM.

The following figure supplement is available for figure 6:

**Figure supplement 1.** Photostimulation of LC terminals in the BLA drive anxiety-like behavior.

## Discussion

We aimed to dissect the functional role of LC to BLA inputs, and to determine the specific receptors within the BLA that mediate LC-NE influenced anxiety-like behavior in the region. Taken together, these data suggest that photostimulation of LC-BLA projections releases NE into the BLA and this

same photostimulation alters BLA firing. This altered firing appears to preferentially increase activity in BLA neurons that project to area known to modulate negative affect such as the CeA and vHPC. In behaving animals, this stimulation of LC terminals in the BLA facilitates a conditioned aversion and acute anxiety-like behaviors. Importantly, the experiments here cannot determine whether the observed changes in BLA unit activity and the conditioned place aversion are mediated directly by NE release from LC terminals. Instead, these affects could be generated through antidromic stimulation of LC cell bodies (and their diverging efferent projections) or through plastic changes resulting from the repeated and long-term stimulation protocols used during the two 30 min conditioning sessions. However, using local, site-specific pharmacological blockade, we identified that $\beta$-ARs within the BLA are necessary for the anxiety-like component of this behavior. These data provide a critical framework for understanding the downstream influence of the LC-NE system in the BLA as it relates to anxiety-like behavioral states.

The BLA is a crucial component in the neural circuitry of negative affective behaviors (*Davis, 1992*; *Davidson, 2002*). Activation of BLA cell bodies and their projections can bidirectionally mediate anxiety-like behavior and fear encoding (*Davis, 1992*; *Kim et al., 2013*; *Tye et al., 2011*; *Namburi et al., 2015*; *Beyeler et al., 2016*; *Wolff et al., 2014*; *Knoll et al., 2011*; *Crowley et al., 2016*; *Sears et al., 2013*; *Roozendaal et al., 2006*; *Felix-Ortiz et al., 2013*; *Debiec and Ledoux, 2004*), but how this behavioral control is altered by neuromodulators is not well understood. NE is increased in the BLA following stressful events (*Galvez et al., 1996*; *Hatfield et al., 1999*). The BLA receives noradrenergic input from the LC (*Asan, 1998*; *Fallon et al., 1978*) as well the *Hoxb1*[+] neuronal populations in the lower brainstem and NTS (*Robertson et al., 2016*). Despite the LC input being less dense than other NE input (*Plummer et al., 2015*), here we directly show that molecularly-defined *Th*[+] positive fibers from the LC release catecholamines into the amygdala and alter behavior. Previous studies have demonstrated that noradrenergic tonic cell firing in the LC increases following stressful stimuli (*McCall et al., 2015*; *Abercrombie and Jacobs, 1987a*, *1987b*; *Aston-Jones et al., 1999*; *Mana and Grace, 1997*). We now demonstrate that selective activation of *Th+* LC terminals from these neurons modulates BLA neuronal activity in vivo. Despite a diversity of recorded responses, these findings align with prior studies suggesting neurons that increase firing to LC-BLA photostimulation have a higher probability of being BLA projection neurons, while those that decrease have a higher probability of being BLA interneurons (*Figure 3L*) (*Likhtik et al., 2006*). Therefore, it seems that it is reasonable that LC-BLA modulation might be selectively increasing activity of BLA projection neurons, though further studies using phototagging or imaging BLA neurons is required. Though it is well established that activation of $\beta$-ARs increases BLA activity (*Buffalari and Grace, 2007*; *Huang et al., 1996*; *Pu et al., 2009*), we cannot conclude that this is the only mechanism by which LC-BLA fibers modulate BLA activity. Indeed, it is likely that the diversity of neuronal responses we observed are mediated through different receptors systems (*Buffalari and Grace, 2007*) and/or local or distal polysynaptic recurrent circuitry (*Wolff et al., 2014*; *Rosenkranz and Grace, 2002*, *1999*). Furthermore, while we do not know the genetic identity or projection targets of the BLA neurons from which we recorded, it is possible that NE serves to preferentially shift BLA activity in neurons that favor anxiety-like behavior (i.e. those that project to the ventral hippocampus, prefrontal cortex, or central amygdala) rather those that favor positive affect and anxiolysis (i.e. projections to the nucleus accumbens and bed nucleus of the stria terminalis)(*Kim et al., 2013*; *Stuber et al., 2011*; *Tye et al., 2011*; *Namburi et al., 2015*; *Beyeler et al., 2016*; *Felix-Ortiz et al., 2013*, *2016*; *Felix-Ortiz and Tye, 2014*). We attempted to begin to address this question using a retrograde labeling strategy combined with immunohistochemistry for cFos. Our results suggest that there is selective modulation of BLA projections neurons. The combined retrograde labeling and photostimulation studies indicate that, cFos, a secondary marker for neuronal excitation (*Madabhushi et al., 2015*), is selectively increased in the CeA- and vHPC-projecting BLA neurons (*Figure 4*). Furthermore, our recent reports demonstrate that increasing the activity of CaMKII+ BLA neurons via chemogenetic or optogenetic activation of G$\alpha$s-signaling also produces an anxiogenic-like state, suggesting that excitatory cells possibly mediate the LC-BLA noradrenergic effect (*Siuda et al., 2015a*, *2016*). However, further work will be necessary to understand the cell-type and projection-specific relationship between LC-NE modulation of BLA activity and its ability to drive negative affective behavior through downstream circuits and receptor systems.

Additionally, we show that photostimulation at the site of LC terminals in the BLA conditions aversive behavior. However, it remains to be seen whether this conditioned aversive behavior is

mediated locally in the BLA by NE. This effect may could be due to factors outside of the BLA and distinct from NE release such as antidromic activity at the LC cell bodies or from prolonged changes to LC circuit activity elsewhere that result from long-term, repeated optogenetic stimulation. However, a similar photostimulation paradigm increases anxiety-like behavior that is blocked by local antagonism of $\beta$-ARs in the BLA. It is well established that prolonged NE release in the BLA modulates memory storage through $\beta$-AR-mediated cAMP production and this effect, in turn, is regulated by $\alpha_1$-AR and $\alpha_2$-AR activity (*Galvez et al., 1996*; *Hatfield et al., 1999*; *Liang et al., 1990*; *Ferry and McGaugh, 2008*; *Ferry et al., 1999a*, *1999b*). In particular, many groups have shown noradrenergic influence in fear-related memory formation (*Sears et al., 2013*; *Roozendaal et al., 2008*, *2006*; *Debiec and Ledoux, 2004*; *Garrido Zinn et al., 2016*; *Skelly et al., 2017*). Our findings support these prior works while offering new insight into how this circuitry might acutely modulate negative affect. The observation that the LC-BLA circuitry carries an intrinsically negative valence suggests that this circuit can natively promote negative affect, rather than merely enhancing memory. However, it is possible that without explicit memory cues, our photostimulation paradigm conditions an aversion via recruitment of many other neurotransmitter systems (*Brioni et al., 1989*; *Marsicano et al., 2002*). This might not be the case in the presence of a positive cue or reward where the LC-BLA projection may still simply enhance the memory. Further studies will be necessary to fully evaluate the implications of the apparent negative affect promoted by exogenous stimulation of this circuit. While there is some concern that photostimulation of $Th^{IRES-Cre}$::LC-BLA:ChR2 fibers could cause backpropagating action potentials to LC-NE cell bodies resulting in NE release elsewhere in the brain for the conditioned place aversion, the observation that the anxiety-like phenotype can be blocked with local $\beta$-AR antagonism (*Figure 6G*) suggests there is some degree of circuit isolation in our study. Additionally, the observation that this stimulation does not replicate all of the behavioral phenotypes of cell body activation (*McCall et al., 2015*) (*Figure 5C*; i.e. no RTPA) further supports this interpretation. However, we cannot definitively conclude that the observed CPA (*Figure 5F and G*) is mediated through the same postsynaptic mechanism (i.e. $\beta$-AR activation) in the BLA as the anxiety-like phenotype. To this point, we recently demonstrated that G$\alpha$s signaling (the same signaling mechanism that $\beta$-AR activates) in CaMKII$\alpha$+ BLA neurons is sufficient for anxiogenesis, without driving either a real-time or conditioned place aversion (*Siuda et al., 2015a*, *2016*). Additionally, throughout these experiments we have not directly addressed any potential confounds that arise from long-term expression of exogenous opsins (*Miyashita et al., 2013*; *Ferenczi et al., 2016*; *Warden et al., 2012*). Instead, our behavioral controls (eYFP-expression with photostimulation) are aimed at controlling spurious effects of exogenous protein expression, heating, and light delivery to the BLA (*Yizhar et al., 2011*). Together with our work at LC-NE cell bodies (*McCall et al., 2015*), these studies and the current findings suggest that LC neurons with fibers in the BLA are capable of conditioning aversion, though possibly through other brain regions and possibly independent of $\beta$-AR activation.

The lack of a real-time place aversion is a particularly interesting observation. We previously demonstrated the real-time place aversion elicited by LC-NE cell body activation to be dependent on $\alpha_1$-AR activation (*McCall et al., 2015*). Together these results suggest that both the anatomical substrate and receptor system responsible for this behavior is distinct from the efferent LC-NE processes responsible for noradrenergic mediated anxiety-like behavior. Furthermore, there are important distinctions between the RTPA and the CPA assays. RTPA is essentially an instrumental conditioning paradigm that relies on operant learning of the subject as it engages the environment, while CPA is a passive, classical conditioning paradigm that possibly recruit distinct brain circuitry during the learning process (*Day and Carelli, 2007*). Another consideration is the longer time course of NE transmission and receptor activation compared to fast-acting, small molecule neurotransmitters, such as $\gamma$-Aminobutyric acid (GABA) and glutamate (*Otis and Mody, 1992*; *Isaacson et al., 1993*; *Zoli et al., 1998*; *Szapiro and Barbour, 2007*; *Agnati et al., 2010*; *Palij and Stamford, 1994*; *Courtney and Ford, 2014*). These temporal dynamics, combined with the role of NE on memory formation (*Sears et al., 2013*; *Roozendaal et al., 2008*, *2006*; *Debiec and Ledoux, 2004*; *Sara, 2000*), might confound the RTPA assay that relies on distinct pairing of the transition from one chamber to the other with changes in neural activity. With activation of adrenergic receptors, these effects may have slower time dynamics than what is necessary for the place-dependent operant conditioning scheme – despite what we observed through cell body activation (*McCall et al., 2015*). It is well known, for example, that $\beta$-AR signaling through cAMP-gated calcium channels that will likely

increase firing acutely, but that they also signal through Arrestin-mediated transduction. The latter can lead to long-term changes in gene and protein expression in addition to changes synaptic plasticity and neuronal activity (*Siuda et al., 2015a*). All of these molecular considerations may also play a key role in the behavior we observed. While numerous potential circuit-based targets exist, it seems possible that NE projections to the mesolimbic dopamine system have the potential to mediate instrumental learning including real-time aversive behaviors. The LC sends functional projections to the dopaminergic system and numerous groups have demonstrated bidirectional control of real-time place preference and aversion behaviors within this system (*Siuda et al., 2015b*; *Jennings et al., 2013*; *Lammel et al., 2012*; *Stamatakis and Stuber, 2012*; *Tan et al., 2012*; *van Zessen et al., 2012*; *Isingrini et al., 2016*).

Taken together, our results here and elsewhere (*McCall et al., 2015*; *Siuda et al., 2015a, 2016*) demonstrate that LC-NE downstream influence on negative affective behaviors is an elaborate and complex system likely involving many different postsynaptic receptor systems and anatomical projection targets. Importantly, the LC projection to the BLA is a clear functional efferent target for LC-mediated anxiety-like behavior. These findings have broad implications for our understanding of the mechanisms of anxiety and other negative affective disorders and suggest that further study of these noradrenergic circuits and signaling pathways should greatly advance our current understanding of related psychopathologies.

## Materials and methods

### Animals

Adult (25–35 g) male C57BL/6J (RRID:IMSR_JAX:000664) and $Th^{IRES-Cre}$ (RRID:IMSR_EM:00254) backcrossed to C57BL/6J mice were group-housed, given access to food and water ad libitum and maintained on a 12 hr:12 hr light:dark cycle. All animals were held in a facility in the lab 1 week prior to surgery, post-surgery and throughout the duration of the behavioral assays to minimize stress from transportation and disruption from foot traffic. To determine animal numbers we used G*Power 3 power analysis software (RRID:SCR_013726) (*Faul et al., 2007*; *Charan and Kantharia, 2013*) informed by data from our typical animal usage logs from similar experiments to suggest a priori means an standard deviations for each intended statistical methods (i.e. Student's t-tests, ANOVAs, etc.) or Mead's resource equation to yield between 10 and 20 degrees of freedom for the error component when no prior data was available (*Charan and Kantharia, 2013*; *Festing and Altman, 2002*). Any variation from these approaches was due to behavioral attrition from off-target injections/implants or headcap failures. All procedures were approved by the Animal Care and Use Committee of Washington University and conformed to US National Institutes of Health guidelines.

### Viral preparation

Plasmids encoding pAAV-EF1α-DIO-eYFP [final titer 5 × 1012 vg/ml], pAAV-EF1α-double floxed-hChR2(H134R)-eYFP-WPRE-HGHpA [final titer 2 × 1013 vg/ml], were obtained from Addgene (Addgene.org) originally from the Deisseroth Laboratory at Stanford University. The DNA was amplified with a Maxiprep kit (Promega) and packaged into AAV5 serotyped viruses by the WUSTL Hope Center Viral Core.

### Stereotaxic surgery

Mice were anaesthetized in an induction chamber (4% isoflurane) and placed in a stereotaxic frame (Kopf Instruments, Model 1900) where they were maintained at 1–2% isoflurane throughout the procedure. A craniotomy was performed and mice were injected as follows. For locus coeruleus terminal studies, 500–1000 nl of AAV5-DIO-ChR2 or AAV5-DIO-eYFP was injected unilaterally (with the exception of two animals for the single-unit electrophysiology experiments that were bilateral) into the locus coeruleus at stereotaxic coordinates from bregma: −5.45 mm anterior-posterior (AP), ±1.25 mm medial-lateral (ML), −3.65 mm dorsal-ventral (DV). Mice were then implanted with chronic fiber optic in the BLA at stereotaxic coordinates from bregma: −1.60 mm AP, ±2.90 mm ML, and −4.75 mm DV. For experiments involving local drug infusion into the BLA, a metal cannula (PlasticsOne) was implanted at −1.60 mm AP, ±2.90 mm ML, and −4.25 mm DV and either an internal infusion cannula or fiber optic was placed to extend another 0.5 mm. The fiber optic or cannula implants

were secured using two bone screws (CMA, 743102) and affixed with TitanBond (Horizon Dental Products) and dental cement (Lang Dental)(*McCall et al., 2015*, *2013*). Mice were allowed to recover for at least 6 weeks prior to behavioral testing; this interval permitted optimal AAV expression and Cre recombinase activity. For retrograde studies 400 nl of Flurogold or Retrobeads were injected into the BLA at −1.60 mm AP, ±2.90 mm ML, and −4.75 mm DV. Mice were allowed to recover for one week for fluorogold and two weeks for retrobeads, prior to perfusion for histological examination. Cholera Toxin subunit b AlexaFluor 555 (Invitrogen; CTB-555) was injected using a Neuros Hamilton Synringe (32 gauge, beveled) to the NAc (+1.25 mm AP, ±0.75 mm ML, −4.30 mm DV; 200 nl), CeM (−0.80 mm AP, ±2.35 mm ML, −5.20 mm DV; 150 nL), vHPC (−3.55 mm AP, +3.50 ML, −4.00 mm DV; 250 nL). The beveled syringe was inserted facing caudally for both NAc and vHPC and medially for CeA injections. CTB was prepared fresh for each series of experiments (*Conte et al., 2009*).

## cFos induction

$Th^{IRES-Cre}$::LC-BLA:ChR2 mice were injected with CTB-555 (Invitrogen, Carlsbad, CA) into the NAc, CeM, or vHPC at least 6 weeks following AAV5-DIO-ChR2-eYFP injection into the LC. Mice were then implanted with a chronic fiber optic in the BLA as described above. Mice were allowed to recover for 5 days and habituated to a chamber and fiber optic tether over 2 days. Seven days following CTB injection, 5 Hz photostimulation (10 ms pulses, 10 mW) was delivered to the BLA for 20 min. Mice were then sacrificed for immunohistochemistry 90 min following the onset of photostimulation. Following immunohistochemistry and imaging, images were background subtracted using ImageJ Fiji (RRID:SCR_002285). Cells were manually counted by an experimenter blind to the experimental conditions.

## Immunohistochemistry

Immunohistochemistry was performed as described (*McCall et al., 2015*; *Siuda et al., 2015a*, *2016*). Briefly, mice were anesthetized with pentobarbital and transcardially perfused with ice-cold 4% paraformaldehyde in phosphate buffer (PB). Brains were dissected, post-fixed for 24 hr at 4°C and cryoprotected with solution of 30% sucrose in 0.1M PB at 4°C for at least 24 hr, cut into 30 μm sections and processed for immunostaining. 30 μm brain sections were washed three times in PBS and blocked in PBS containing 0.5% Triton X-100% and 5% normal goat serum. Sections were then incubated for ~16 hr at room temperature in chicken anti-TH (1:2000, Aves Labs, Tigard, OR) or rabbit anti-phospho-cFos (1:500, Cell Signaling Technology, Danvers, MA). Following incubation, sections were washed three times in PBS and then incubated for 2 hr at room temperature goat anti-chicken or anti-rabbit Alexa Fluor 594 or 633 (1:500, Invitrogen, Carlsbad, CA) were then washed three times in PBS and followed by three 10 min rinses in PB and mounted on glass slides with Hardset Vectashield (Vector Labs, Burlingame, CA) (RRID:AB_2336787) for microscopy. All sections were imaged on both epifluorescent and confocal microscopes. Gain and exposure time were constant throughout each experiment, and all image groups were processed in parallel using Adobe Photoshop CS5 (Adobe Systems, San Jose, CA) (RRID:SCR_014199).

| Antibody | Species | Dilution | Source | RRID |
|---|---|---|---|---|
| TH | Chicken | 1:2000 | Aves Labs | RRID:AB_10013440 |
| Alexa Fluor 594 anti-chicken IgG | Goat | 1:500 | Invitrogen | RRID:AB_142803 |
| Alexa Fluor 633 anti-chicken IgG | Goat | 1:500 | Invitrogen | RRID:AB_1500591 |
| Alexa Fluor 633 anti-rabbit IgG | Goat | 1:500 | Invitrogen | RRID:AB_2535731 |
| Phospho-cFos | Rabbit | 1:500 | Cell Signaling | RRID:AB_10557109 |

## Slice preparation

Slice electrophysiology and voltammetry experiments were performed as previously described (*Stamatakis et al., 2013*). Briefly, mice were anesthetized (Euthasol) and perfused with ice-cold sucrose aCSF (in mM: 225 sucrose, 119 NaCl, 1.0 NaH2PO4, 4.9 MgCl2, 0.1 CaCl2, 26.2 NaHCo3,

1.25 glucose). Following rapid decapitation, coronal slices containing LC and BLA were prepared on a vibratome (VT-1200, Leica Microsystems, Wetzlar, Germany) and allowed to rest for at least 30 min in oxygenated aCSF (in mM: 119 NaCl, 2.5 KCl, 1.0 NaH2PO4, 1.3 MgCl2, 2.5 CaCl2, 26.2 NaHCO3, and 11 glucose) at 35°C prior to placement in the recording chamber where slices were perfused with oxygenated aCSF at 37°C at a rate of 2 ml/min.

## Slice electrophysiology

Whole-cell current clamp recordings were made in LC neurons expressing ChR2-eYFP with a Multiclamp 700B amplifier (Molecular Devices, Sunnyvale, CA) (RRID:SCR_011323). Since LC neurons are spontaneously active, current was injected such that the cell was resting at −65 mV. To elicit action potentials, slices were stimulated with 5 pulses of blue LED at a rate of 10 Hz (Thorlabs, 473 nm, 5 ms pulse width, 1 mW) via the 40X objective.

## Fast scan cyclic voltammetry

Carbon fiber microelectrodes (CFME, 75 µm in length) were lowered into 300 µM coronal slices and placed where the densest ChR2-eYFP expression was observed (CeA/BLA border) $Th^{IRES-Cre}$::LC-BLA:ChR2 mice. To detect NE, the CFME was ramped from −0.4 V to 1.3 V versus a Ag/AgCl reference electrode (in the bath) at a rate of 400 V/s at 10 Hz. Slices were stimulated with 30 pulses of a blue LED (Thorlabs, 473 nm, 5 ms pulse width, 1 mW) via a 40X objective at 10 Hz every 5 min to release NE. Electrochemical data was collected and analyzed using a combination of Tar Heel CV (ESA, Chelmsford, MA) (*Robinson and Wightman, 2007*), HDCV (http://www.chem.unc.edu/facilities/electronics_software.html) (*Bucher et al., 2013*), and Labview (RRID:SCR_014325). Following collection, background subtracted cyclic voltammograms (CVs) were smoothed one time with a Fast Fourier Transformation (*Bucher et al., 2013*). CVs had characteristic oxidation and reduction peaks coinciding with catecholamine detection (ox: 600–700 mV red: −200–300 mV). Oxidative currents were analyzed at the peak of the oxidative potential for individual experiments. Clearance half-life (t1/2) was measured in Clampfit 10.2 (Molecular devices) as previously described (*McElligott et al., 2013*).

## In vivo electrophysiology

Spontaneous single unit activity was recorded as previously described (*McCall et al., 2015*; *Siuda et al., 2015a*). Briefly, mice were lightly anesthetized (1% isoflurane), placed in a stereotactic frame and two skull screws were placed on either side of the midline to ground the electrode array. The recording apparatus consisted of a 16-channel (35 µm tungsten wires, 150 µm spacing between wires, 150–33 µm spacing between rows, Innovative Physiology, Durham, NC) electrode array. This array was epoxied to a fiber optic and lowered into the BLA (stereotaxic coordinates from bregma: −1.3 mm (AP),±2.9 mm (ML) and −4.9 mm (DV). Extracellular recordings were taken from $Th^{IRES-Cre}$:: LC-BLA:ChR2 mice. Spontaneous and photostimulated neuronal activity was recorded from each electrode, bandpass-filtered with activity between 250 and 8000 Hz, and analyzed as spikes. Voltage signals were amplified and digitally converted using OmniPlex and PlexControl (Plexon, Dallas, TX) (RRID:SCR_014803). For $Th^{IRES-Cre}$::LC-BLA:ChR2 recordings, 3 min of baseline recordings were made followed by 3 min of 5 Hz photostimulation (10 ms pulses, 10 mW) and then another 3 min post- stimulation (off, on, off). Principle component analysis and/or evaluation of t-distribution with expectation maximization was used to sort spikes using Offline Sorter (Plexon)(RRID:SCR_000012) and only cells with distinct clusters away from the noise that remained firing throughout the duration of the recording were included. To assess firing rate changes for each cell, all spikes were binned into 1 s bins. Using z-scores ($z = (x-\mu)/\sigma$; where x is the sampled firing rate of the neuron per bin), firing rates were normalized to the mean baseline firing rate during the 60 s before the start of the photostimulation. Neurons were classified as increasing firing during photostimulation if their average z-score during photostimulation was greater than 1. Similarly, neurons were classified as decreasing firing if their average z-score during photostimulation was less than −1. All units with average z-scores from −1 to 1 during photostimulation were classified as no change. To determine the latency to fire, we calculated the average time from the onset of first photostimulation to the next spike from each cell, independent of whether the cell classified as increasing, decreasing, or static. To determine baseline firing rate, all spikes were binned into 1 s bins and the mean of the first

180 s of the recording (prior to any photostimulation) was used. Average waveforms were computed from each isolated unit using the first 540 s of each recording session. Subsequently, we performed linear correlations between each neuron of a designated class (i.e. increasing, decreasing, or static) where r = 1 would be an identical waveform. Bursts were defined as at least two spikes with an inter-spike interval (ISI) of <170 ms to start a burst, and two spikes with an ISI > 300 ms to terminate a burst. Percent spikes in bursts was quantified by recording the number of spikes in bursts out of the total number of spikes from a unit.

## General behavior notes

Behavioral assays were performed in a sound attenuated room maintained at 23°C. Lighting was measured and stabilized at ~4 lux for anxiety tests and ~200 lux for place testing. All behavioral apparatuses were cleaned with 70% ethanol in between animals. In each assay, animals received constant photostimulation throughout the entire trial. AAV5-DIO-ChR2 and AAV5-DIO-eYFP animals received 5 ms pulses of 10 Hz photostimulation (473 nm). Movements were video recorded and analyzed using Ethovision Software (RRID:SCR_000441).

## Real time place testing (RTPT)

Mice were placed into a custom-made unbiased, balanced two-compartment conditioning apparatus (52.5 × 25.5 × 25.5 cm) as previously described (*McCall et al., 2015*; *Siuda et al., 2015b*; *Al-Hasani et al., 2015*) and allowed to freely roam the entire apparatus for 20 min. Entry into one compartment triggered constant photostimulation (5 Hz for LC-BLA; ~10 mW light power) while the animal remained in the light-paired chamber. Entry into the other chamber ended the photostimulation. The side paired with photostimulation was counterbalanced across mice. Time spent in each chamber and total distance traveled for the entire 20 min trial was measured using Ethovision 8.5 (Noldus, Wageningen, Netherlands). Data are expressed as mean ± S.E.M percent time spent in photostimulation-paired chamber.

## Conditioned place aversion (CPA)

Mice were trained in an unbiased, balanced three compartment conditioning apparatus as previously described (*McCall et al., 2015*). Briefly, mice were pre-tested by placing individual animals in the small central compartment and allowing them to explore the entire apparatus for 30 min. Time spent in each compartment was recorded with a video camera (ZR90; Canon, Tokyo, Japan) and analyzed using Ethovision 8.5 (Noldus). Mice were randomly assigned to light and no-light compartments and received no light in the morning and light (5 Hz for LC-BLA) in the afternoon at least 4 hr after the morning training on two consecutive days. CPA was assessed on day 4 by allowing the mice to roam freely in all three compartments and recording the time spent in each. Scores were calculated by subtracting the time spent in the light stimulus-paired compartment post-test minus the pre-test.

## Open field test (OFT)

OFT testing was performed as described (*McCall et al., 2015*) in a square enclosure (50 × 50 cm). We connected $Th^{IRES-Cre}$::LC-BLA:ChR2 mice to fiber optics and allowed them to roam freely for 21 min. Photostimulation alternated between off and on (5 Hz (10 ms width)) photostimulation (~10 mW light power)) states in 3 min time segments, beginning with 3 min of no stimulation. The open field was cleaned with 70% ethanol between each trial. The center was defined as a square comprised of 25% the total area of the OFT (i.e. each length was 50% that of the total OFT).

## Elevated zero maze (EZM)

EZM testing was performed as described (*McCall et al., 2015*), the EZM (Harvard Apparatus, Holliston, MA) was made of grey plastic, 200 cm in circumference, comprised of four 50 cm sections (two opened and two closed). The maze was elevated 50 cm above the floor and had a path width of 4 cm with a 0.5 cm lip on each open section. $Th^{IRES-Cre}$::LC-BLA:ChR2 animals were connected to cables coupled to a function generator, positioned head first into a closed arm, and allowed to roam freely for 7 min. Animals received 5 Hz (10 ms width) photostimulation (~10 mW light power). For the $\beta$-AR antagonism experiment, mice were injected into the BLA with Propranolol

(1 µg, intra-BLA, 30 min prior to behavior, Tocris, Bristol, United Kingdom). Mean open arm time was the primary measure of anxiety-like behavior.

## Genotyping of mouse lines

DNA was isolated from tail tissue obtained from weanling mice (21–28 days of age), and PCR screening was performed using the following primers: Cre recombinase (forward: 5'- GCA TTA CCG GTC GAT GCA ACG AGT GAT GAG-3' and reverse: 5'- GAG TGA ACG AAC CTG GTC GAA ATC AGT GCG-3') yielding a 400 bp PCR product in Cre positive animals. Fatty acid-binding protein intestinal primers (forward: 5'- TGG ACA GGA CTG GAC CTC TGC TTT CCT AGA-3' and reverse: 5'- TAG AGC TTT GCC ACA TCA CAG GTC ATT CAG-3') were used as positive controls and yield a 200 bp PCR product.

## Statistics/data analysis

All summary data are expressed as mean ± SEM. Statistical significance was taken as *p<0.05, **p<0.01, ***p<0.001, as determined by the Student's t-test (paired and unpaired): One-Way Analysis of Variance (ANOVA) or One-Way Repeated Measures ANOVA, followed by Dunnett's or Bonferroni post hoc tests as appropriate. In cases where data failed the D'Agostino and Pearson omnibus normality test, non-parametric analyses were used. Statistical analyses were performed in GraphPad Prism 5.0 (RRID:SCR_002798).

# Acknowledgements

This work is supported by NIMH R01 MH112355 (MRB), and early work in this study by NIDA R21 DA035144 (MRB), the McDonnell Center for Systems Neuroscience (MRB), NIMH F31 MH101956 (JGM), WUSTL DBBS (JGM, ERS), start-up funds from Washington University in St. Louis (JGM), NIAAA K01 AA023555 (ZAM), and the Alcohol Beverage Medical Research Foundation (ZAM). We thank all of the Bruchas Laboratory for their helpful insight and discussion throughout the preparation of the manuscript. We also thank Thomas J Baranski, N Gautam, Robert W Gereau, IV, Erik D Herzog, Timothy E Holy, Joseph D Dougherty, and Joe Henry Steinbach for helpful discussions. We thank Karl Deisseroth (Stanford University) for the channelrhodopsin-2 (H134) construct. We also thank The WUSTL HOPE Center viral vector core for related viral preparations (NINDS, P30NS057105).

# Additional information

## Competing interests

MRB: Michael R. Bruchas, PhD is a co-founder of Neurolux, Inc, a company that is making wireless optogenetic and various neuroscience-related probes. None of the work in this manuscript used these devices or is related to any of the company's activities, but we list this information here in full disclosure. The other authors declare that no competing interests exist.

## Funding

| Funder | Grant reference number | Author |
|---|---|---|
| National Institute of Mental Health | MH101956 | Jordan G McCall |
| Washington University in St. Louis | | Jordan G McCall Edward R Siuda |
| National Institute on Alcohol Abuse and Alcoholism | AA023555 | Zoe A McElligott |
| Alcohol Beverage Medical Research Foundation | | Zoe A McElligott |
| National Institute on Drug Abuse | DA032750 | Garret D Stuber |
| National Institute on Drug Abuse | DA035144 | Michael R Bruchas |

| McDonnell Center for Systems Neuroscience | | Michael R Bruchas |
| National Institute of Mental Health | MH112355 | Michael R Bruchas |

The funders had no role in study design, data collection and interpretation, or the decision to submit the work for publication.

## Author contributions

JGM, ERS, Conceptualization, Data curation, Formal analysis, Investigation, Visualization, Methodology, Writing—original draft, Writing—review and editing; DLB, Data curation, Investigation, Visualization, Methodology, Writing—review and editing; LAL, Investigation; ZAM, Investigation, Methodology, Writing—review and editing; GDS, Supervision, Methodology; MRB, Conceptualization, Funding acquisition, Methodology, Writing—original draft, Project administration, Writing—review and editing

## Author ORCIDs

Jordan G McCall, http://orcid.org/0000-0001-8295-0664
Dionnet L Bhatti, http://orcid.org/0000-0003-3031-2067
Zoe A McElligott, http://orcid.org/0000-0002-4717-5698
Garret D Stuber, http://orcid.org/0000-0003-1730-4855
Michael R Bruchas, http://orcid.org/0000-0003-4713-7816

## Ethics

Animal experimentation: This study was performed in strict accordance with the recommendations in the Guide for the Care and Use of Laboratory Animals of the National Institutes of Health. All of the animals were handled according to approved institutional animal care and use committee (IACUC) protocols at Washington University in St. Louis. The protocol was approved by the Animal Studies Committee at Washington University in St. Louis (Protocol Number: 20130219; expiration date: 15/10/2016). All surgery was performed under isoflurane anesthesia, and every effort was made to minimize suffering.

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
