## [Decision Letter]

Thank you for submitting your article "Activation of locus coeruleus to basolateral amygdala projections drives anxiety and aversion" for consideration by *eLife*. Your article has been favorably evaluated by a Senior Editor and three reviewers, one of whom is a member of our Board of Reviewing Editors. The following individual involved in review of your submission has agreed to reveal her identity: Kay M Tye (Reviewer #2).

As you see from the reviews below, all reviewers are enthusiastic about the topic and findings, and each has raised a number of recommendations for revision. After discussion among all three reviewers, we would like to invite you to submit a revised manuscript that can satisfactorily address the following:

1) We feel that the in vivo recording data in Figure 3 need to be significantly strengthened. Specifically, "n" should be increased to have more examples of up- or down-modulated BLA neurons. It will be ideal if in these experiments you can learn more about the recorded cells' other properties, such as their response to natural stimuli and modulations by propranolol, but we do not hold these as essential for the acceptance for a revised manuscript.

2) We would also request that you perform the propranolol experiment in the aversion assay (Figure 4), which will strengthen your claim that the effect on aversion is mediated by LC-NE neuron projection to BLA rather than via antidromic stimulation elsewhere. If this additional data cannot be obtained, you should significantly tone down the claim that aversion is related to LC-NE modulation of BLA, including in the title.

3) We believe that most of the other critiques can be addressed by further analyses of existing data, and by textual revisions. You should try to address these as thoroughly as possible.

*Reviewer #1:*

McCall et al. characterized the action of locus coeruleus norepinephrine (LC-NE) neurons at one of their numerous target sites: the basolateral amygdala (BLA), in the context of anxiety and aversion. They confirmed previous findings that LC-NE neurons project and arborize axons at BLA, showed that photoactivation of ChR2-expressing LC-NE neurons resulted in NE release at BLA as measured by fast scan cyclic voltammetry, characterized firing pattern changes of BLA neurons in response to photostimulation of LC-NE neurons using in vivo recording, found that terminal stimulation of LC-NE fibers at BLA caused conditioned (but not real time) place aversion and enhanced anxiety-like behaviors, and showed that the anxiety-like behaviors can be suppressed by application of propranolol, an antagonist of β-adrenergic receptor.

Overall this is a nice and timely set of experiments addressing an important question: how do LC-NE neurons affect the physiology and function of a specific target area, in this case BLA that is well known for its function in regulating emotion and anxiety. Some of the experiments and analyses could go further.

1) The in vivo recording (Figure 3) of BLA neurons in response to photostimulation of LC-NE neurons is perhaps the least satisfactory experiment. I will start with a few questions about the description. There were 20 cells in Figure 3, but only 18 were further analyzed in Figure 3 (and onwards?). What were the selection criteria? How was "latency" measured (Figure 3) for cells that had no change in firing rate? Do Figure 3 contain new cells (legend for 3M says n=10), or a subset of cells in Figure 3 or Figure 3? Again, how are these cells selected?

Given the clear heterogeneity of response, it will be more satisfying if the authors have some more knowledge about these cells-for example are there any signatures that distinguish three types of neurons in the PCA analysis? Do they respond differently to natural stimuli? Do they respond differently to propranolol, which will provide a nice link to Figure 5? Increasing N is desirable as the examples are only 4 and 2 for neurons whose firing rate is up- or down-modulated by LC-NE neuron photostimulation.

2) Given the broad projection patterns of LC-NE neurons, the behavioral effect of LC-NE photostimulation in Figure 4 and Figure 5 could in principle be due to antidromic effects in other target areas. The propranolol effect caused by local application in Figure 5 supports a BLA local action. The propranolol experiment should be extended to Figure 4 as well.

*Reviewer #2:*

Overall, I was delighted to read this interesting and important manuscript. I have relatively minor comments.

I found it intriguing that the authors observed such heterogeneous responses to LC-BLA photostimulation and wonder if the differences in responses of BLA neurons to NE is related to either a) their projection target, b) cell type (projection or interneuron) or c) their role during anxiety-related behaviors. This is beyond the scope of the current manuscript and I do not expect the authors to respond to this query with further experiments, but if the authors have any other data regarding basal spike rate, waveform, peak firing rates, burst patterns or activity during anxiety behaviors it would be nice to see. The authors must have the firing profile data already, and just need to analyze and display this.

The Authors state that "Downstream and independent of this projection, recent studies have demonstrated that direct activation of both basolateral amygdala (BLA) cell bodies or their projections is both anxiogenic and socially aversive" and should consider adding the following papers to the references already listed: Felix-Ortiz, Neuron, 2013; Felix-Ortiz, J Neurosci, 2014; Felix-Ortiz, Neuroscience, 2015. These citations should also be added to the references following the sentence "Furthermore, while we do not know the genetic identity or projection targets of the BLA neurons from which we recorded, it is possible that NE serves to preferentially shift BLA activity in neurons that favor anxiety-like behavior (i.e. those that project to the ventral hippocampus, prefrontal cortex, or central amygdala) rather those that favor positive affect and anxiolysis (i.e. projections to the nucleus accumbens and bed nucleus of the stria terminalis)".

The finding that LC-BLA inputs drive conditioned place aversion potentially place previous studies concluding that NE in the BLA contributes to fear memory formation/storage. The authors should discuss this body of research and clearly address how their data may challenge this existing notion. Specifically, the finding that this circuit is intrinsically of negative valence may actually provide an alternative explanation for a number of results previously thought to support a role in memory formation (associated with fear). The authors discuss these results as being consistent, though, it's not clear to me that this is the case.

It was curious that they were not able to produce real-time place aversion but were able to observe conditioned place aversion. The authors interpret this as an issue of the involvement of long-term memory. An alternative interpretation would be that this effect is specific to Pavlovian and not instrumental conditioning. Real-time place aversion is explicitly different in that the animal must enter the paired side of the chamber to receive photostimulation. Thus, a number of other issues may also be involved. First, this is essentially an instrumental conditioning task, as the photostimulation is contingent on animal behavior. Second, this may be related to the lingering effects of NE (as the authors hint at, but don't directly discuss).

The appropriate control for the propranolol experiment would be a saline/vehicle injection (both in combination with ChR2 stimulation), as well as a light-off condition wherein ChR2 is still expressed. ChR2 expression can (with long-term expression) change basal behavior even in the absence of light (see Warden et al., Nature, 2012). It seems as if they may have done this based on the supplementary figure, but the main text was confusing. Please clarify.

In Figure 1, what is above the ventricle? There is some retrograde labeling from the BLA that is TH-. Also, it is hard to see if this exists in D, but it seems to. Finally, some quantification of the TH overlap with cre-dependent fluorophore expression should be included. Fibers in BLA are scarcely visible in 1L, I am surprised this level of expression was sufficient to evoke behavioral responses – was this image representative?

In Figure 3, the firing rate appears to be dropping prior to photostimulation – which makes it unclear whether the effect we are observing is actually related to the optogenetic manipulation.

*Reviewer #3:*

Major conclusions of paper:

McCall et al. combine in vivo optogenetics, physiology, pharmacology, and behavior, as well as FSCV, to characterize a specific sub-circuit of the extensive LC-NE system. Their experiments identify novel examples of heterogeneity within the LC→ BLA circuit, substantially build on existing work in the field, and have important implications for understanding how NE modulates anxiety/aversion. While interesting, I felt that some of their conclusions regarding specificity within this circuit were not experimentally demonstrated as strong as they could be, but hopefully, this can be addressed during revision. I also had some technical questions regarding experiments for Figure 3.

Main concerns:

Figure 3:

1) Sample size seems low for statistical analysis (4 cells in increased firing group, 2 cells in decreased group).

2) I was confused by 3C. Is each dot a cell? How do they correspond to the pie chart in 3D?

3) It says in the Materials and methods that injections were unilateral, but according to Figure 3 legend, data collection was from "4 hemispheres from 2 animals". Almost all LC projections to forebrain are ipsilateral (though I'm not sure about projections to BLA), so it doesn't seem possible that both hemispheres could be used for these experiments if only one LC was injected.

4) I'm unsure if the statistical comparisons are appropriate; i.e. – pre-sorting of the "increased" and "decreased" cells based on average firing.

5) It would be nice to expand on the identity of the NE-responding BLA neurons. What if adrenergic inhibitors are applied to the BLA during their optrode recordings? How does this affect the heterogeneous NE-mediated activity they observed? Alternatively, could CRACM-style experiments be performed in slice, where recordings are performed in BLA neurons when ChR2-expressing LC fibers are stimulated? This could provide higher resolution for the relative size of these different NE-responding populations, as well as their identity (via post-recording cell fill and immunolabeling?)

Regarding optogenetic stimulation of fibers/related to Figure 4: The authors find that stim. of BLA→ LC fibers produces negative association over time, which isn't present in real-time place aversion. But is it known if avoidance association is specific to the BLA→LC projection? It also seems possible that chronic activation of LC terminals could cause a change in LC cell body activity, and thus broadly affect the LC-NE circuit. This is mentioned in the Discussion, but maybe it's possible to experimentally address this concern. For instance, does local infusion of an adrenergic receptor antagonist into the BLA during or after conditioning block the avoidance (similar to the experiment done in Figure 5)?

---

## [Author Response]

*As you see from the reviews below, all reviewers are enthusiastic about the topic and findings, and each has raised a number of recommendations for revision. After discussion among all three reviewers, we would like to invite you to submit a revised manuscript that can satisfactorily address the following:*

*1) We feel that the in vivo recording data in Figure 3 need to be significantly strengthened. Specifically, "n" should be increased to have more examples of up- or down-modulated BLA neurons. It will be ideal if in these experiments you can learn more about the recorded cells' other properties, such as their response to natural stimuli and modulations by propranolol, but we do not hold these as essential for the acceptance for a revised manuscript.*

We appreciate the concerns regarding Figure 3 and agree that these experiments needed bolstering. Due to the above cited consideration on mouse number we were unable to perform all the suggested experiments, but we were able to more than double the n from the original submission and perform the important request for expanded analyses on this data. While were unable to test the responses of these cells to natural stimuli, we were able to follow-up with a functional anatomy experiment that provides some clarity to the projection targets of the modulated cells. In particular, we used retrograde labeling combined with cFos staining (a secondary marker for neuronal activation) to identify that LC-BLA stimulation appears to bias increased activity to BLA circuits known to promote aniety-like behavior (i.e. the central amygdala and the ventral hippocampus). Together, we hope these new data are more comprehensive and well powered than what was presented in the initial submission.

*2) We would also request that you perform the propranolol experiment in the aversion assay (Figure 4), which will strengthen your claim that the effect on aversion is mediated by LC-NE neuron projection to BLA rather than via antidromic stimulation elsewhere. If this additional data cannot be obtained, you should significantly tone down the claim that aversion is related to LC-NE modulation of BLA, including in the title.*

In this particular case of the conditioned place aversion experiments, we were unable to perform the necessary experiments to fully control for effects outside of the BLA from antidromic stimulation due to limited mice expand the in vivo recordings, which were indicated by reviewers and review summary as the top priority for the revision. We do agree that optogenetic manipulations are terminal sites come with many caveats of interpretation and we have taken every effort to tone down our conclusions regarding these experiments and have therefore removed this “aversion” concept from the title of the article. More details are included below.

*3) We believe that most of the other critiques can be addressed by further analyses of existing data, and by textual revisions. You should try to address these as thoroughly as possible.*

We appreciate these other concerns raised and have tried to address these directly and completely. More details are included below.

*Reviewer #1:*

*[…] 1) The in vivo recording (Figure 3) of BLA neurons in response to photostimulation of LC-NE neurons is perhaps the least satisfactory experiment. I will start with a few questions about the description. There were 20 cells in Figure 3, but only 18 were further analyzed in Figure 3 (and onwards?).*

We appreciate this attention to detail by the reviewer and apologize for the mistake. Figure 3 includes two cells firing at a low rate that reached a firing rate zero for the remainder of the recording session such that we judged them to be lost, from the electrode rather than simply not firing. They should have not been included in 3C in the original submission and have been removed from this submission.

*What were the selection criteria?*

We have adjusted the wording in the Materials and methods section to be more explicitly clear as to how single cells were selected. It now reads, “Principle component analysis and/or evaluation of t-distribution with expectation maximization was used to sort spikes using Offline Sorter (Plexon) and only cells with distinct clusters away from the noise that remained firing throughout the duration of the recording were included.”

*How was "latency" measured (Figure 3) for cells that had no change in firing rate?*

We regret that this was not clearer in the initial submission. The latency presented here is simply the time to the first spike following the first photostimulation. Therefore a ‘latency’ exists for all included cells. We have added this explanation to the Materials and methods section which reads: “To determine the latency to fire, we calculated the average time from the onset of first photostimulation to the next spike from each cell, independent of whether the cell classified as increasing, decreasing, or static.”

*Do Figure 3 contain new cells (legend for 3M says n=10), or a subset of cells in Figure 3 or Figure 3? Again, how are these cells selected?*

These cells represent a subset of 3C/3D where the recording was judged sufficiently stable to continue recording for an additional manipulation. These recordings were done prior to the analysis to determine whether each cell increased or decreased (or stayed the same) to the 5 Hz stimulation. To be clearer, we have altered the text to read: “Furthermore, in a subset of cells blindly selected (without knowledge of the increase/decrease/static response to photostimulation) following the 5 Hz recordings we also observed similar increases in firing rates to constant photostimulation, where the overall population of neurons increased firing during stimulation (Figure 3—figure supplement 1).”

*Given the clear heterogeneity of response, it will be more satisfying if the authors have some more knowledge about these cells-for example are there any signatures that distinguish three types of neurons in the PCA analysis? Do they respond differently to natural stimuli? Do they respond differently to propranolol, which will provide a nice link to Figure 5? Increasing N is desirable as the examples are only 4 and 2 for neurons whose firing rate is up- or down-modulated by LC-NE neuron photostimulation.*

We appreciate the concerns regarding Figure 3 and agree that these experiments needed bolstering. Due to the above cited consideration on mouse number we were unable to perform all of the suggested experiments, however we were able to more than double the n of the original submission and perform the requested expanded analyses on this larger data set. Notably, this doubling preserved the ratio of up- and down-modulated cells almost exactly as the smaller n. We also examined more properties of the existing and newly acquired cells, including sorting them by baseline firing rate, determining the similarity of the waveforms within each group, and quantifying their bursting properties. In the case of sorting the neurons by baseline firing, we can now observe that up-modulated cells have a lower baseline firing rate than down-modulated cells. This finding suggests that it may be more likely that these up-modulated cells are BLA projection neurons, rather than interneurons (Likhtik et al., Journal of Neurophysiology, 2006). However, this is only a likelihood and further experiments would be necessary to fully draw this conclusion. We have adjusted the figures and text accordingly to reflect these observations.

While were unable to test the responses of these cells to natural stimuli, we did expand the study with a follow-up functional anatomy experiment that provides some clarity to the projection targets of the modulated cells. In particular, we used retrograde labeling combined with cFos staining to identify that LC-BLA stimulation appears to bias increased activity to BLA circuits known to promote aniety-like behavior (i.e. the central amygdala and the ventral hippocampus). Together, we think these new data are more satisfying than what was presented in the initial submission.

*2) Given the broad projection patterns of LC-NE neurons, the behavioral effect of LC-NE photostimulation in Figure 4 and Figure 5 could in principle be due to antidromic effects in other target areas. The propranolol effect caused by local application in Figure 5 supports a BLA local action. The propranolol experiment should be extended to Figure 4 as well.*

We agree with the reviewer that this is a critical experiment for the interpretation of the aversion experiments. Unfortunately, we were not able to secure sufficient TH-IRES-Cre mouse cohorts on a reasonable timeline to complete these experiments properly with all controls and implants (~40+ Cre+ animals). Instead, we have taken every effort to tone down our conclusions throughout the paper regarding the aversion experiments in (the original) Figure 4 and have also appropriately removed aversion from the title of the paper, focusing instead on the observed anxiety-like behaviors. Future experiments (including those to eliminate the confound of backpropagating action potentials) will be necessary to determine the precise role of LC-NE fibers in the BLA for aversive behaviors.

*Reviewer #2:*

*Overall, I was delighted to read this interesting and important manuscript. I have relatively minor comments.*

*I found it intriguing that the authors observed such heterogeneous responses to LC-BLA photostimulation and wonder if the differences in responses of BLA neurons to NE is related to either a) their projection target, b) cell type (projection or interneuron) or c) their role during anxiety-related behaviors. This is beyond the scope of the current manuscript and I do not expect the authors to respond to this query with further experiments, but if the authors have any other data regarding basal spike rate, waveform, peak firing rates, burst patterns or activity during anxiety behaviors it would be nice to see. The authors must have the firing profile data already, and just need to analyze and display this.*

We agree that these suggestions are key components to better understanding the system. We have taken two approaches to addressing these concerns. First, we added a new experiment and figure to the manuscript attempting to address, as a first pass, whether BLA neurons that increase in activity following LC-BLA photostimulation are segregated by projection target. To do so, we retrogradely labeled BLA neurons with a fluorophore-tagged Cholera Toxin B in either the ventral hippocampus, central amygdala, or nucleus accumbens, stimulated LC-BLA fibers, and observed co-localization between the CTB label and the presence of cfos, a secondary marker for neuronal activation. These studies indicate a bias towards activation of BLA neurons that project to ventral hippocampus and central amygdala, rather than the nucleus accumbens. While these experiments suggest that LC-BLA biases activation towards these projection targets that are known to promote anxiety-like behavior, much more work will be necessary in subsequent studies to clearly elucidate the intricacies of these tri-synaptic circuits. In addition to these new experiments, we have also added more recordings (as discussed elsewhere) and replotted our findings based on the baseline firing profile of these neurons, their average waveform, and we have quantified their bursting properties by group as well. As stated previously, the sorting the neurons by baseline firing demonstrates that cells that increase in response to the photostimulation have a lower baseline firing rate than down-modulated cells. This finding suggests that it may be more likely that these up-modulated cells are BLA projection neurons, rather than interneurons (Likhtik et al., Journal of Neurophysiology, 2006). However, this is only a likelihood and further experiments would be necessary to fully draw this conclusion. We have adjusted the figures and text accordingly to reflect these observations.

*The Authors state that "Downstream and independent of this projection, recent studies have demonstrated that direct activation of both basolateral amygdala (BLA) cell bodies or their projections is both anxiogenic and socially aversive" and should consider adding the following papers to the references already listed: Felix-Ortiz, Neuron, 2013; Felix-Ortiz, J Neurosci, 2014; Felix-Ortiz, Neuroscience, 2015. These citations should also be added to the references following the sentence "Furthermore, while we do not know the genetic identity or projection targets of the BLA neurons from which we recorded, it is possible that NE serves to preferentially shift BLA activity in neurons that favor anxiety-like behavior (i.e. those that project to the ventral hippocampus, prefrontal cortex, or central amygdala) rather those that favor positive affect and anxiolysis (i.e. projections to the nucleus accumbens and bed nucleus of the stria terminalis)".*

We appreciate this attention to detail by the reviewer. While we had already cited Felix-Ortiz, Neuron, 2013 and Felix-Ortiz, Neuroscience, 2015, we had previously failed to cite Felix-Ortiz, J Neurosci, 2014. That citation is now included in both locations of the manuscript.

*The finding that LC-BLA inputs drive conditioned place aversion potentially place previous studies concluding that NE in the BLA contributes to fear memory formation/storage. The authors should discuss this body of research and clearly address how their data may challenge this existing notion. Specifically, the finding that this circuit is intrinsically of negative valence may actually provide an alternative explanation for a number of results previously thought to support a role in memory formation (associated with fear). The authors discuss these results as being consistent, though, it's not clear to me that this is the case.*

This interpretation is an important consideration we did not highlight it clearly enough in our original submission. We have altered the text to provide more detail on the existing body of research as well as elaborating on our intended interpretation. The new text reads: “It is well established that prolonged NE release in the BLA modulates memory storage through β-AR-mediated cAMP production and this effect, in turn, is regulated by α_1_-AR and α_2_-AR activity (Debiec and Ledoux, 2004; Galves, Mesches and McGaugh, 1996; Madabhushi et al., 2015; Liang, McGaugh and Yao, 1990; Ferry and McGaugh, 2008; Ferry, Roozendaal and McGaugh, 1999). […] Further studies will be necessary to fully evaluate the implications of the apparent negative affect promoted by exogenous stimulation of this circuit.”

*It was curious that they were not able to produce real-time place aversion but were able to observe conditioned place aversion. The authors interpret this as an issue of the involvement of long-term memory. An alternative interpretation would be that this effect is specific to Pavlovian and not instrumental conditioning. Real-time place aversion is explicitly different in that the animal must enter the paired side of the chamber to receive photostimulation. Thus, a number of other issues may also be involved. First, this is essentially an instrumental conditioning task, as the photostimulation is contingent on animal behavior. Second, this may be related to the lingering effects of NE (as the authors hint at, but don't directly discuss).*

These are importantalternative considerations and we have integrated them into the Discussion as such: “Furthermore, there are important distinctions between the RTPA and the CPA assays. […] All of these molecular considerations may also play a key role in the behavior we observed.”

*The appropriate control for the propranolol experiment would be a saline/vehicle injection (both in combination with ChR2 stimulation), as well as a light-off condition wherein ChR2 is still expressed. ChR2 expression can (with long-term expression) change basal behavior even in the absence of light (see Warden et al., Nature, 2012). It seems as if they may have done this based on the supplementary figure, but the main text was confusing. Please clarify.*

We apologize that this was not clearer in the initial submission. The EZM experiment had four groups eYFP+Vehicle, ChR2+Vehicle, eYFP+Propranolol, and ChR2+ Propranolol. This approach allows comparisons to detect a change evoked from photostimulation (eYFP+Vehicle vs. ChR2+Vehicle) as well as an effect from the pharmacological treatment (ChR2+Vehicle vs. ChR2+ Propranolol) while controlling for any effects of the antagonist independent of photostimulation (eYFP+Propranolol vs. ChR2+ Propranolol). We have revised the Results section to more clearly state these details, it now reads: “To do so we implanted a combined fiber-optic fluid cannula into the BLA and delivered either an artificial cerebrospinal fluid vehicle or the non-selective β-AR antagonist, Propranolol (1 µg)(Sears et al., 2013; Roozendaal et al., 2006; Buffalari and Grace, 2009; Buffalari and Grace, 2007), prior to photostimulation (Figure 6, Figure 6—figure supplement 1). […] Importantly, when we locally antagonized β-ARs directly in the BLA prior to photostimulation, anxiogenesis was completely blocked in ChR2^+^ animals with no effect on eYFP-expressing, propranolol controls (Figure 6).”

Furthermore, we do agree that controls for the long-term expression of ChR2 are important, however we were unfortunately unable to incorporate these experiments into this submission due to limited animal numbers. We have addressed this limitation now in the Discussion, stating: “Additionally, throughout these experiments we have not directly addressed any potential confounds that arise from long-term expression of exogenous opsins (Galves, Mesches and McGaugh, 1996; Hatfield, Spanis and McGaugh, 1999; Likhtik et al., 2006). Instead, our behavioral controls (eYFP-expression with photostimulation) are aimed at controlling spurious effects of exogenous protein expression, heating, and light delivery to the BLA (74).”

*In Figure 1, what is above the ventricle? There is some retrograde labeling from the BLA that is TH-. Also, it is hard to see if this exists in D, but it seems to.*

It seems as though the initial image/labeling on the image caused confusion. There is nothing above the ventricle (labeled 4V on image). However, there is retrograde labeling to TH- cells that are dorsal and ventral to the LC itself (top right of the image in 1B). These cells are likely part of the medial parabrachial nucleus which have previously identified projections to the BLA (Saper & Loewy, Brain Research, 1980). We have made a note regarding this labeling in the Figure 1 legend.

*Finally, some quantification of the TH overlap with cre-dependent fluorophore expression should be included.*

We agree that using any Cre-driver line for cell-type selective modulation should be well documented for its selectivity. For this concern, we point the reviewer to our previous manuscript where we indeed included a detailed description of this overlap for noradrenergic, DBH^+^ cells. In particular, the data presented in Figures S2C, 5E & S5A of McCall et al., Neuron, 2015 directly addresses this question by comparing viral expression to dopamine β hydroxylase (DBH) staining with essentially 100% of Cre-dependent viral expression overlapping with DBH^+^ cells. We therefore felt it was redundant to do this again.

*Fibers in BLA are scarcely visible in 1L, I am surprised this level of expression was sufficient to evoke behavioral responses – was this image representative?*

We appreciate this attention to detail by the reviewer. The images used throughout the manuscript were selected to be representative of what is typically seen. We have included the data from the Allen Institute (now Figure 1—figure supplement 1) to support these observations as well. We see similar expression in the BLA as has been observed elsewhere (e.g. Plummer et al., Development, 2015). We agree that it is remarkable that such diffuse innervation has meaningful behavioral effects, but believe this to be a characteristic feature of the noradrenergic system through its known volume transmission properties (e.g. Courtney & Ford, Journal of Neuroscience, 2014). A further important consideration for neuromodulation, acting via metabotropic receptors is that density of inputs, fluorescence, or fiber labeling do not necessarily predict how well coupling amplification can happen after a GPCR is activated. For example, in many cases a receptor pool of say 100 receptors, only needs to be occupied by 10% by norepinephrine to have a maximal possible effect on a neuronal response. This concept, termed ‘spare receptors’ is because of large amplification that happens after the G-protein coupling event, cAMP generation, and channel modulations. So, while we appreciate this is a visually weak input, there are numerous other examples in both the central and peripheral nervous system where modulatory inputs (including by NE) by visualization alone look weak, yet these fibers carry and amplify robust messages that dramatically effect behavior and circuits. This is also as an aside, why interpreting viral tracing density visually as a measure of function can be a confound in some cases.

*In Figure 3, the firing rate appears to be dropping prior to photostimulation – which makes it unclear whether the effect we are observing is actually related to the optogenetic manipulation.*

We appreciate this observation by the reviewer, with additional recordings we no longer see this ‘anticipatory’ decrease in firing. We do note, however, that while the photostimulation-induced increase in firing returns to baseline quickly following the end of the photostimulation, the suppression of firing appears to be maintained long after stimulation ends.

*Reviewer #3:*

*Major conclusions of paper:*

*McCall et al. combine in vivo optogenetics, physiology, pharmacology, and behavior, as well as FSCV, to characterize a specific sub-circuit of the extensive LC-NE system. Their experiments identify novel examples of heterogeneity within the LC→ BLA circuit, substantially build on existing work in the field, and have important implications for understanding how NE modulates anxiety/aversion. While interesting, I felt that some of their conclusions regarding specificity within this circuit were not experimentally demonstrated as strong as they could be, but hopefully, this can be addressed during revision. I also had some technical questions regarding experiments for Figure 3.*

We appreciate this positive assessment of our work and hope that you will find this revised manuscript a stronger contribution to the field.

*Main concerns:*

*Figure 3:*

*1) Sample size seems low for statistical analysis (4 cells in increased firing group, 2 cells in decreased group).*

We appreciate concerns that the initial data set was too limited. We were able to now more than double the n of the original submission and perform the requested expanded analyses on this data. Notably, this doubling preserved the ratio of increasing and decreasing cells almost exactly as the smaller n.

2) I was confused by 3C. Is each dot a cell? How do they correspond to the pie chart in 3D?

We apologize for any confusion as originally presented. We have altered the figure to connect the dots from pre- and during stimulation in 3C. For clarity, we have also segregated the cells into increasing, decreasing, and static and have presented these in separate graphs in the new Figure 3—figure supplement 1. We hope this expanded view is clearer.

*3) It says in the Materials and methods that injections were unilateral, but according to Figure 3 legend, data collection was from "4 hemispheres from 2 animals". Almost all LC projections to forebrain are ipsilateral (though I'm not sure about projections to BLA), so it doesn't seem possible that both hemispheres could be used for these experiments if only one LC was injected.*

We appreciate this attention to detail and apologize for the minor error in reporting. All animals were injected unilaterally except for the two used for single unit electrophysiology the initial submission. The new animals recorded and reported in this submission were all injected unilaterally and recorded from the ipsilateral BLA. We have noted these two exceptions in the Materials and methods, stating: “For locus coeruleus terminal studies, 500-1000 nl of AAV5-DIO-ChR2 or AAV5-DIO-eYFP was injected unilaterally (with the exception of two animals for the single-unit electrophysiology experiments that were bilateral) into the locus coeruleus at stereotaxic coordinates from bregma: -5.45 mm anterior-posterior (AP), ± 1.25 mm medial-lateral (ML), -3.65 mm dorsal-ventral (DV).”

*4) I'm unsure if the statistical comparisons are appropriate; i.e. – pre-sorting of the "increased" and "decreased" cells based on average firing.*

We agree that statistical comparisons based on pre-sorted groups are inappropriate. Based on editorial comments from a presubmission inquiry, we removed any direct statistical comparison based on the firing rate of neurons (the feature used to classify the cells). Instead we employed the z-score method that we (Jennings et al., Nature, 2013) and others (e.g. Wolff et al., Nature, 2014) have used previously to identify gross changes in neuronal activity. The inclusion of statistics for the basal frequency measure is still appropriate as this feature was not part the classification scheme and the descriptive statistics for latencies remain for informative purposes. All other statistics in this figure and its supplement are done on the whole sample without any pre-classification.

*5) It would be nice to expand on the identity of the NE-responding BLA neurons. What if adrenergic inhibitors are applied to the BLA during their optrode recordings? How does this affect the heterogeneous NE-mediated activity they observed? Alternatively, could CRACM-style experiments be performed in slice, where recordings are performed in BLA neurons when ChR2-expressing LC fibers are stimulated? This could provide higher resolution for the relative size of these different NE-responding populations, as well as their identity (via post-recording cell fill and immunolabeling?)*

We appreciated these elegant suggestions to improve our work. Unfortunately, most of these suggestions were outside of the scope of what was possible for this study given the struggles we had with animal breeding. While the adrenergic inhibitors would be an important addition to this story, the practical limitations of recording in anesthetized mice meant that we would need many fold more animals than we had available after exhausting all possible sources/supplies. With the pressing need to increase our n for the recording experiments, we focused efforts on acquiring that new data with the hopes of following up with adrenergic compounds, however, we were unable to add sufficient numbers of recordings with the pharmacological treatments, so no conclusion could be made and these are not included in the manuscript. We do, however, believe that the new figure detailing LC-BLA photostimulation-induced increases in cfos to be a first step in answering some of the concerns raised here. While it lacks the temporal precision that would be provided by the suggested CRACM-style slice recordings, the new experiment suggests that there is projection selectivity in the modulation of BLA activity by the LC. Future work will endeavor to answer these questions more completely.

*Regarding optogenetic stimulation of fibers/related to Figure 4: The authors find that stim. of BLA→ LC fibers produces negative association over time, which isn't present in real-time place aversion. But is it known if avoidance association is specific to the BLA→LC projection? It also seems possible that chronic activation of LC terminals could cause a change in LC cell body activity, and thus broadly affect the LC-NE circuit. This is mentioned in the Discussion, but maybe it's possible to experimentally address this concern. For instance, does local infusion of an adrenergic receptor antagonist into the BLA during or after conditioning block the avoidance (similar to the experiment done in Figure 5)?*

As previously discussed, we agree with the reviewer that this is a critical experiment for the interpretation of the aversion experiments. Unfortunately, we were not able to secure sufficient TH-IRES-Cre mice on a reasonable timeline to complete these experiments. Instead, we have taken every effort to tone down our conclusions throughout the paper regarding the experiments in (the original) Figure 4 and have removed aversion from the title of the paper, focusing instead on the observed anxiety-like behaviors. Future experiments (including those to eliminate the confounding effect of potential backpropagating action potentials) will be necessary to determine the precise role of LC-NE fibers in the BLA for aversive behaviors.